# Cost-Sensitive Freeze-thaw Bayesian Optimization for Efficient Hyperparameter Tuning

**Dong Bok Lee[1]**    **Aoxuan Silvia Zhang[1]***    **Byungjoo Kim[1]***    **Junhyeon Park[1]***
**Steven Adriaensen[2]**    **Juho Lee[1]**    **Sung Ju Hwang[1,3]**    **Hae Beom Lee[4]**

[1]KAIST    [2]University of Freiburg    [3]DeepAuto.ai    [4]Korea University

markhi@kaist.ac.kr    haebeomlee@korea.ac.kr

## Abstract

In this paper, we address the problem of *cost-sensitive* hyperparameter optimization (HPO) built upon freeze-thaw Bayesian optimization (BO). Specifically, we assume a scenario where users want to early-stop the HPO process when the expected performance improvement is not satisfactory with respect to the additional computational cost. Motivated by this scenario, we introduce *utility* in the freeze-thaw framework, a function describing the trade-off between the cost and performance that can be estimated from the user's preference data. This utility function, combined with our novel acquisition function and stopping criterion, allows us to dynamically continue training the configuration that we expect to maximally improve the utility in the future, and also automatically stop the HPO process around the maximum utility. Further, we improve the sample efficiency of existing freeze-thaw methods with transfer learning to develop a specialized surrogate model for the cost-sensitive HPO problem. We validate our algorithm on established multi-fidelity HPO benchmarks and show that it outperforms all the previous freeze-thaw BO and transfer-BO baselines we consider, while achieving a significantly better trade-off between the cost and performance. Our code is publicly available at https://github.com/db-Lee/CFBO.

## 1    Introduction

Hyperparameter optimization [HPO; 9, 22, 8, 48, 16, 34, 12] stands as a crucial challenge in the domain of deep learning, given its importance in achieving optimal empirical performance. Unfortunately, the field of HPO for deep learning remains relatively underexplored, with many practitioners resorting to simple trial-and-error methods [8, 34]. Moreover, traditional black-box Bayesian optimization (BO) approaches for HPO [9, 48, 12] face limitations when applied to deep neural networks due to the impracticality of evaluating a vast number of hyperparameter configurations until convergence, each of which may take several days.

Recently, multi-fidelity HPO [53, 34, 14, 4, 62, 3, 24, 44] has gained increasing attention to improve the sample efficiency of traditional black-box HPO. It leverages lower-fidelity information (*e.g.*, validation accuracies at fewer training epochs) to predict and optimize performance at higher or full fidelity (*e.g.*, validation accuracies at the last training epoch). Furthermore, unlike black-box HPO, multi-fidelity HPO dynamically selects hyperparameter configurations even before finishing a single training run, demonstrating its ability of finding better configurations sample-efficiently.

---

*Equal Contribution.

39th Conference on Neural Information Processing Systems (NeurIPS 2025).

However, one critical limitation of the conventional multi-fidelity HPO frameworks is their lack of awareness of the *trade-off between the cost and performance*. For example, given a limited amount of total credits, customers of cloud computing services (*e.g.*, GCP, AWS, or Azure) can heavily penalize the cost of HPO relative to its performance to conserve credits for other tasks. A similar scenario applies to task manager users such as Slurm, who aim to optimize their allocated time within a computing instance. In such cases, users may prefer that the HPO process focuses on exploiting the current belief about good hyperparameter configurations rather than exploring new ones, to efficiently consume their limited resources. However, existing methods [53, 34, 14, 4, 62, 3, 24, 44] generally do not consider this scenario, as they typically assume a sufficiently large budget (*e.g.*, total credits or allocated time) and aim to achieve the best performance on a validation set.

**Utility: trade-off between cost and performance.**
Therefore, in this paper, we introduce a more sophisticated notion of cost sensitivity for HPO. Specifically, we assume that users have their own preferences regarding the trade-off between the cost and performance of HPO. We formalize this trade-off in a *utility* function that represents user preferences and can be estimated from user preference data. It assigns higher values as costs decrease and performance increases, and vice versa. Some users may strongly penalize the budget spent on HPO, while others may penalize it weakly or not at all, as in the conventional multi-fidelity HPO. We explicitly maximize this utility by dynamically selecting hyperparameter configurations expected to achieve the greatest improvement in the future and by automatically

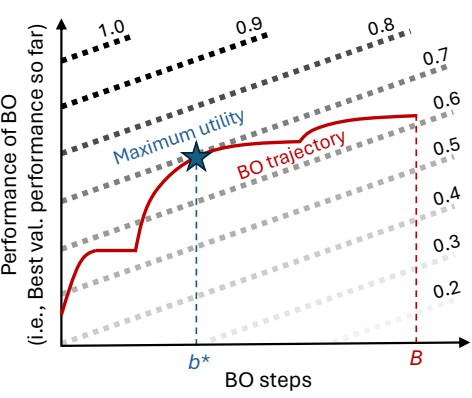

Figure 1: A concept of **user utility**.

terminating the BO around the maximum utility instead of halting at an arbitrary target budget.

Fig. 1 illustrates the concept of utility: the red line shows the BO trajectory, and the blue asterisk marks the maximum utility (around 0.7) achieved at budget $b^*$. This example heavily penalizes the additional cost, although extending the BO to the full budget $B$ could yield better performance.

Solving this problem requires our method to have the following capabilities. Firstly, it should support **freeze-thaw BO** [53, 44], an advanced form of multi-fidelity BO that dynamically pauses (freezes) and resumes (thaws) hyperparameter configuration trainings based on future performance extrapolated from a set of partially observed learning curves (LCs) with various configurations. Such efficient and fine-grained allocation of computational resources aligns well with the goal of achieving the best trade-off between cost and performance in multi-fidelity HPO. Secondly, freeze-thaw BO requires that its surrogate function be capable of **LC extrapolation** [24, 2, 44]. In our case, this is critical for making probabilistic inferences about future utilities, which guide the selection of the best configuration and enable precise early stopping of the HPO. Lastly, since users are assumed to prefer stopping HPO as early as possible when performance saturates, LC extrapolation must be accurate even in the early stages of HPO. Therefore, it is essential to use **transfer learning** to maximize the sample efficiency of BO [3] and to prevent premature stopping.

Based on these criteria, we introduce our novel **C**ost-sensitive **F**reeze-thaw **BO** (**CFBO**), which effectively maximizes utility using the three components mentioned above. We first explain the notation and background[2] on freeze-thaw BO and Prior-Fitted Networks [PFNs; 39, 2, 44] for LC extrapolation (§2). We define the utility function and explain how to estimate it from user preference data[3] (§3.1). We describe the acquisition function and the stopping criterion specifically developed for our problem setting, explaining how they achieve a good trade-off between cost and performance (§3.2 and §3.3). We show how to train a PFN using existing real-world LC datasets to create a sample-efficient in-context surrogate function for freeze-thaw BO, which effectively captures correlations between different hyperparameter configurations (§3.4). Finally, we empirically demonstrate the superiority of CFBO on a diverse set of utility functions, three multi-fidelity HPO benchmarks, and a real-world object detection LC dataset we collected, showing that it significantly outperforms relevant baselines, including transfer-BO (§4).

---

[2]We defer the discussion of related work, *e.g.*, early-stopping BO methods, to Appendix A.

[3]Some users may already have an exact form of their utility function, but for others, we provide a method to quantify it based on their preference data.

We summarize our contributions and findings as follows:

- We propose a new problem formulation, cost-sensitive multi-fidelity HPO, which focuses on maximizing the trade-off between cost and performance (*i.e.*, utility) as defined by users, rather than optimizing asymptotic validation performance.

- We introduce a novel acquisition function and stopping criterion specifically designed for this problem formulation, incorporating transfer learning to enhance in-context LC extrapolation.

- We extensively validate the effectiveness of our method across diverse cost-sensitive multi-fidelity HPO scenarios using three popular LC benchmarks.

## 2 Background and Related Work

**Notation.** Let $\mathcal{X} = \{x_1, \ldots, x_N\}$ denote the set of hyperparameter configurations, where $x_n \in \mathbb{R}^{d_x}$ and $N$ is the number of configurations [62, 24, 44]. Let $t \in [T] := \{1, \ldots, T\}$ denote the training epochs, with $T$ being the last epoch, and $y_{n,1:T} := (y_{n,1}, \ldots, y_{n,T}) \in [0, 1]^T$ the measure of model performance to be maximized, *e.g.*, a learning curve (LC) of validation accuracies with $x_n$. We now introduce the notation for multi-fidelity BO. $t_n < T$ denotes the last observed epoch of $x_n$ if the model performance of $x_n$ is partially observed ($y_{n,1:t_n}$). Let $B$ denote the total budget spent during BO, and $\tilde{y}_b \in [0, 1]$ the best cumulative performance achieved up to budget $b \in [B]$, respectively. See Tab. 4 in Appendix B for a summary of the notation used throughout this paper.

**Freeze-thaw BO.** Freeze-thaw BO [53] is an advanced form of multi-fidelity BO, which aims to maximize the best cumulative performance $\tilde{y}_B$ by efficiently allocating the limited total budget $B$. Assuming that we allocate one budget unit (*i.e.*, one epoch) for each freeze-thaw BO step, it allows us to dynamically select and evaluate the best hyperparameter configuration $x_{n^*}$, with $n^* \in [N]$ denoting the corresponding index, while pausing the evaluation of the previously selected configuration. Specifically, given the context $\mathcal{C} = \{(x, t, y)\}$, which represents a set of partial (or full) LCs collected up to a specific BO step (*i.e.*, *history*), we select one configuration $x_{n^*} \in \mathcal{X}$ that maximizes a predefined acquisition function (*e.g.*, the expected improvement [38]). We then evaluate the selected configuration $x_{n^*}$ for one additional budget unit (*i.e.*, one epoch) and observe $y_{n^*, t_{n^*}+1}$. Next, we update the history $\mathcal{C}$ with $(x_{n^*}, t_{n^*} + 1, y_{n^*, t_{n^*}+1})$. The cumulative best validation performance $\tilde{y}_b$ is updated accordingly. This process is repeated for $B$ steps. Please see Alg. 1 for the pseudocode (except for the blue parts).

**PFNs for LC extrapolation.** Freeze-thaw BO usually requires the ability to extrapolate LCs to compute acquisition functions [62, 24, 44]. Among many plausible options, in this paper, we use Prior-data Fitted Networks [PFNs; 39] for the LC extrapolation. PFNs are an in-context Bayesian inference method based on Transformer [54] and show good performance in LC extrapolation [2, 44] without computationally expensive online retraining [62, 24]. Specifically, after training, they allow us to approximate the posterior predictive distribution (PPD) with a single forward pass: $p_\theta(y|x, t, \mathcal{C}) \approx p(y|x, t, \mathcal{C})$, where $p_\theta$ is the approximate PPD parameterized by $\theta$. This is achieved by minimizing the cross-entropy for predicting the hold-out example's label $y$, given $x$, $t$, and $\mathcal{C}$:

$$\mathcal{L}(\theta) = \mathbb{E}_{(x,t,y),\mathcal{C} \sim p(\mathcal{D})} \left[ -\log p_\theta(y|x, t, \mathcal{C}) \right], \tag{1}$$

where $p(\mathcal{D})$ is a prior from which we can sample infinitely many *synthetic* training data. We defer the architectural and training details to Appendix E.

**Related work.** We defer the discussion of related work on (1) **multi-fidelity HPO**, (2) **freeze-thaw BO**, (3) **LC extrapolation**, (4) **transfer-BO**, (5) **cost-sensitive HPO**, (6) **early stopping BO**, (7) **BO with user preferences**, and (8) **neural processes** to Appendix A.

## 3 Method: Cost-sensitive Freeze-thaw Bayesian Optimization (CFBO)

We now introduce our method, Cost-sensitive Freeze-Thaw Bayesian Optimization (CFBO), summarized in Alg. 1, where the blue components indicate the parts specific to our approach.

### 3.1 Utility: Trade-off between Cost and Performance

**Utility function.** A utility function $U : (b, \tilde{y}_b) \in [B] \times [0, 1] \mapsto [0, 1]$ describes the trade-off between the budget $b$ and the best cumulative performance $\tilde{y}_b$. Its value decreases with increasing

$b$ and increases with increasing $\tilde{y}_b$. We assume that the utility is given by the users. For example, it can be simply defined as $U(b, \tilde{y}_b) = \tilde{y}_b - \alpha \left(\frac{b}{B}\right)^c$, where $0 \leq \alpha \leq 1$ is a penalty coefficient and $c = 1, 2,$ or $0.5$ for a linear, quadratic, or square-root utility, respectively. Furthermore, the total budget limit can be modeled by setting $U(b, \tilde{y}_b) = -\infty$ if $b > B$ otherwise $\tilde{y}_b$.

**Utility estimation.** It is often challenging for users to quantify their preference on the trade-off. We therefore propose to use the Bradley-Terry model [10] for utility estimation:

$$p(U(b, \tilde{y}_b) > U(b', \tilde{y}'_{b'})) = \frac{\exp(U(b, \tilde{y}_b)/\tau)}{\exp(U(b, \tilde{y}_b)/\tau) + \exp(U(b', \tilde{y}'_{b'})/\tau)}. \tag{2}$$

where $U$ is the user utility we want to estimate, and $\tau$ is a temperature hyperparameter. Eq. 2 describes the probability that the user prefers $(b, \tilde{y}_b)$ to $(b', \tilde{y}'_{b'})$ in terms of utility $U$. Following the preference learning literature [6], we can collect *user preference data* by asking users which point they prefer: (1) sampling a pair of points $(b, \tilde{y}_b), (b', \tilde{y}'_{b'})$, (2) labeling user preference label $y_> \in \{0, 1\}$ as 1 if $U(b, \tilde{y}_b) > U(b', \tilde{y}'_{b'})$ otherwise 0, and (3) constructing a dataset $\mathcal{D}_U := \{((b, \tilde{y}_b), (b', \tilde{y}'_{b'}), y_>)\}$. We then optimize the parameter of the utility function (*e.g.*, the penalty coefficient $\alpha$) by minimizing the binary cross-entropy loss $\ell(x, y) := -y \log x - (1-y) \log(1-x)$:

$$\frac{1}{|\mathcal{D}_U|} \sum_{(b, \tilde{y}_b),(b', \tilde{y}'_{b'}), y_> \in \mathcal{D}_U} \ell\big(p\left(U\left(b, \tilde{y}_b\right) > U\left(b', \tilde{y}'_{b'}\right)\right), y_>\big). \tag{3}$$

**An example of utility estimation.** Fig. 2 illustrates an example of our utility estimation process. Here, we assume a simulated user who consistently favors outcomes that improve upon the baseline freeze-thaw HPO trajectory, *e.g.*, ifBO [44]. We first run ifBO for up to 300 budgets and obtain $\forall b \in [B], \tilde{y}_b$, where $\tilde{y}_b$ denotes the best cumulative performance of BO up to budget $b$. For each $b \in \{51, \ldots, 300\}$, we then sample $\tilde{y}_b^{(\text{up})} \sim$ Uniform$(\tilde{y}_b, 1)$ and $\tilde{y}_b^{(\text{down})} \sim$ Uniform$(0, \tilde{y}_b)$, and record the preference $U(b, \tilde{y}_b^{(\text{up})}) > U(b, \tilde{y}_b^{(\text{down})})$. We exclude the initial budgets $b \in \{1, \ldots, 50\}$ to prevent the utility model from overfitting to

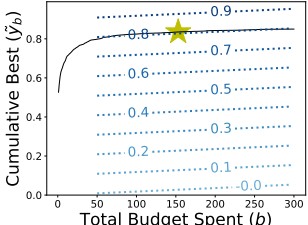

Figure 2: An example of **utility function estimation**.

the early steep improvements. This simulates a user who *always prefers* outcomes above the baseline trajectory. The solid black line in Fig. 2 represents the ifBO trajectory ($\tilde{y}_b$), the dotted blue lines indicate the estimated utility, and the yellow asterisk marks the maximum utility. We observe that the utility reaches its maximum when the improvement becomes *saturated*.

**In practice.** We recommend providing a small library of functional forms—*e.g.*, linear ($c = 1$), quadratic ($c = 2$), square-root ($c = 0.5$), or staircase—and fitting the utility via preference learning using Eqs. 2 and 3. Assuming the true utility of users follows one of these forms, Appendix C shows that it can be recovered with only a few pairwise comparisons (*e.g.*, 30 pairs).

## 3.2 Acquisition

**EI-based acquisition function.** We define the acquisition function $A(n; U)$ for maximizing utility based on the Expected Improvement (EI) method [38]:

$$A(n; U) := \max_{\Delta t \in [T - t_n]} \mathbb{E}_{y_{n,\cdot} \sim p_\theta} \left[ [U(b + \Delta t, \tilde{y}_{b + \Delta t}) - U_{\text{p}}]^+ \right], \tag{4}$$

where $[x]^+ := \max(x, 0)$. In Eq. 4, we first extrapolate $y_{n,\cdot} := \{y_{n, t_n + \Delta t} \mid \Delta t \in [T - t_n]\}$, the remaining part of the learning curve (LC) associated with $x_n$, using $p_\theta$. Then, we compute the corresponding cumulative best performances $\{\tilde{y}_{b + \Delta t} \mid \Delta t \in [T - t_n]\}$. According to the definition in §2,

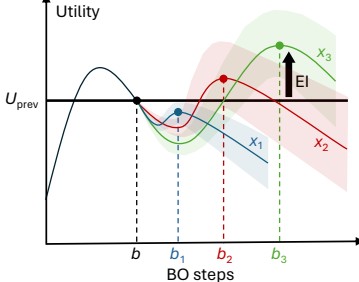

Figure 3: An illustration of **A in Eq. 4**.

$\tilde{y}_{b + \Delta t}$ is computed by taking the maximum of the previous best performance $\tilde{y}_b$ and the newly extrapolated validation performances $y_{n, t_n + 1}, \ldots, y_{n, t_n + T}$. Based on the updated budget $b + \Delta t$ and the corresponding performance $\tilde{y}_{b + \Delta t}$, we compute the utility and its expected improvement over

the previous utility $U_{\mathrm{p}}$. The expectation is evaluated over the distribution of LC extrapolations ($p_\theta$), using Monte Carlo estimation (detailed in Appendix F). Finally, the acquisition $A(n; U)$ for each configuration index $n$ is determined by selecting the best increment $\Delta t \in [T - t_n]$ that maximizes the expected improvement. We then choose the best index over configurations $n^*$ that maximizes $A(n; U)$, *i.e.*, $n^* = \arg\max_n A(n; U)$. Assuming three configurations $x_1, x_2$, and $x_3$, Fig. 3 illustrates the selection of configuration $x_3$.

**Differences from existing EI.** The main differences between our acquisition function in Eq. 4 and the usual EI-based acquisition are twofold. First, instead of maximizing the expected improvement of the validation performance $y$, we maximize the *EI of utility*. Second, rather than fixing the target epoch for evaluating the acquisition to the last epoch $T$ [24] or using a random increment [44], we dynamically select the *best target epoch* that is expected to yield the highest improvement in utility.

These aspects enable our BO framework to carefully select configurations at each suggestion step, aiming to achieve the best trade-off between the cost and performance of the HPO process. Specifically, the acquisition function initially favors configurations that are expected to yield strong asymptotic validation performances $y$. However, as the BO progresses, the acquisition function gradually becomes greedier as the performance saturates and the associated cost begins to dominate the utility function (empirically illustrated in Figs. 8a to 8c of §4.3). Consequently, the acquisition function shifts from exploration to exploitation—prioritizing the current configurations over selecting new ones to maximize short-term performance.

---

**Algorithm 1** Cost-Sensitive Freeze-thaw BO.
Blue parts correspond to specifics of our method.

---

1: **Input:** $\mathcal{X}$ : hyperparameter configuration space
2:         $p_\theta$ : LC extrapolator with transfer learning
3:         $A$ : acquisition function
4:         $B$ : total HPO budget
5:         $U$ : utility function
6:         $\delta$ or $\beta, \gamma$ : threshold hyperparameters
7: $\tilde{y}_0 \leftarrow -\infty,\ \mathcal{C} \leftarrow \emptyset,\ t_1, \ldots, t_N \leftarrow 0, U_{\mathrm{p}} \leftarrow 0,\ \hat{R}_b \leftarrow 0$
8: **for** $b = 1, \ldots, B$ **do**
9:      $n^* \leftarrow \arg\max_n A(n; U)$      ▷ Acquisition in Eq. 7
10:      $\delta_b \leftarrow \mathrm{BetaCDF}(p_b; \beta, \beta)^\gamma$     ▷ $p_b$ in Eq. 4
11:      **if** $\hat{R}_b > \delta$ or $\delta_b$ **then**
12:         **break**        ▷ Stopping criterion
13:      **end if**
14:      Evaluate $y_{n^*, t_{n^*}+1}$ with $x_{n^*}$
15:      $\mathcal{C} \leftarrow \mathcal{C} \cup \{(x_{n^*}, t_{n^*}+1, y_{n^*, t_{n^*}+1})\}$
16:      $\tilde{y}_b \leftarrow \max(\tilde{y}_{b-1}, y_{n^*, t_{n^*}+1})$
17:      $t_{n^*} \leftarrow t_{n^*} + 1$
18:      $U_{\mathrm{p}} \leftarrow U(b, \tilde{y}_b)$
19:      $\hat{R}_b \leftarrow \frac{\hat{U}_{\max} - U_{\mathrm{p}}}{\hat{U}_{\max} - \hat{U}_{\min}}$    ▷ Estimated regret in Eq. 5
20: **end for**
21: **Output:** The model trained with $x^*$ up to the $t^*$-th epoch s.t. $(x^*, t^*, y^*) \coloneqq \arg\max_{(x,t,y) \in \mathcal{C}} y$

---

**Utility is irreversible.** Note that $U_{\mathrm{p}}$ in Eq. 4, the reference value for EI, is *not* the greatest utility achieved so far (corresponds to $f^*$ in the typical EI $A(x) \coloneqq \mathbb{E}_f[f - f^*]^+$). Instead, it is set to the utility value achieved most recently (*i.e.*, $U_p$ in line 18 of Alg. 1), as the computational budgets spent are *irreversible*. This also contrasts with typical EI-based BO settings, where all previous evaluations remain meaningful, allowing the reference value can be set to the maximum among them. Consequently, $U_{\mathrm{p}}$ can either increase or decrease during the BO process.

### 3.3 Stopping Criterion

**Regret-based criterion.** The next question is how to properly stop the HPO around the maximum utility. We propose stopping when the following criterion is satisfied:

$$\hat{R}_b := \frac{\hat{U}_{\max} - U_{\mathrm{p}}}{\hat{U}_{\max} - \hat{U}_{\min}} > \delta. \qquad (5)$$

In Eq. 5, $U_{\mathrm{p}}$ is the utility value at the previous step $b-1$, $\hat{U}_{\max}$ is defined as the maximum utility value seen up to the previous $(b-1)$ step, and $\hat{U}_{\min} = U(B, \tilde{y}_1)$. The role of $\hat{U}_{\max}$ and $\hat{U}_{\min}$ is to *roughly estimate* the maximum and minimum utility achievable over the course of BO, respectively. $\hat{R}_b \in [0, 1]$ can be seen as roughly estimated *normalized regret* at the current step $b$. This regret-based criterion terminates the BO process once the estimated regret exceeds a predefined threshold $\delta \in [0, 1]$. The intuition is *pessimistic*: if the current utility regret is already high, it is unlikely that further optimization will yield a significant improvement, and thus the BO process is terminated.

**Adaptive threshold.** We can fix the threshold $\delta$ as a hyperparameter in Eq. 5 (*e.g.*, baselines); however, this approach does not account for *possibility* of potential improvement in the future. To

address this, we propose an *adaptive threshold* based on the probability of improvement [**PI**; 38]:

$$\delta_b = \text{BetaCDF}(p_b; \beta, \beta)^\gamma, \quad \beta, \gamma > 0, \tag{6}$$

$$p_b = \max_{\Delta t \in [T - t_n]} \mathbb{E}_{y_{n^*, \cdot} \sim p_\theta} \left[ \mathbb{1} \left( U \left( b + \Delta t, \tilde{y}_{b + \Delta t} \right) > U_{\text{p}} \right) \right]. \tag{7}$$

Here, $\text{BetaCDF}$ is the cumulative distribution function (CDF) of the $\text{Beta}$ distribution, and $\mathbb{1}$ is the indicator function. The PI $p_b$ in Eq. 7 represents the probability that the selected configuration $x_{n^*}$ using Eq. 4 improves $U_{\text{p}}$ in some future BO step. Intuitively, we aim to defer termination as $p_b$ increases and vice versa. This behavior is incorporated into Eq. 6—as $p_b$ increases, the adaptive threshold $\delta_b$ also increases because $\text{BetaCDF}(\cdot; \beta, \beta)^\gamma$ is a monotonically increasing function in $[0, 1]$. Consequently, according to Eq. 5, there is less motivation to terminate the BO process when $p_b$ is high. For our CFBO, we use the adaptive threshold $\delta_b$ instead of the fixed threshold $\delta$ in Eq. 5.

**Role of $\gamma$ and $\beta$.** Fig. 4a shows that $\gamma$ controls $\delta_b$ at $p_b = 0.5$. For example, if we set $\gamma = \log_{0.5} 0.2$, $\delta_b$ becomes 0.2 when $p_b = 0.5$ regardless of $\beta$. By controlling $\gamma$, we can set the threshold to a proper value when the PI is uncertain ($p_b = 0.5$). Fig. 4b illustrates the effect of $\beta$. As $\beta \to 0$, $\delta_b$ becomes horizontal, fixing $\delta_b$ at 0.2 regardless of $p_b$. This results in **ignoring the PI** $p_b$. In contrast, as $\beta \to \infty$, $\delta_b$ becomes vertical, causing $\delta_b$ to take binary values, either 0 or 1, depending on whether $p_b > 0.5$ or not. This corresponds to a **purely PI-based criterion**. Thus, $\beta$ provides a smooth interpolation that controls the degree to which the PI $p_b$ in Eq. 7 is incorporated into the adaptive threshold $\delta_b$ in Eq. 6 and the stopping criterion in Eq. 5.

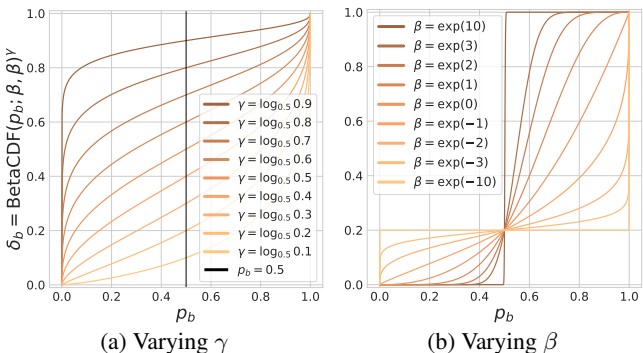

Figure 4: **(a)** $\delta_b$ vs. $p_b$ for **varying $\gamma$** with $\beta = \exp(-1)$. **(b)** $\delta_b$ vs. $p_b$ for **varying $\beta$** with $\gamma = \log_{0.5} 0.2$.

## 3.4 Transfer Learning with LC mixup

**Motivation.** Since users may want to early-stop the BO process, it is crucial to ensure an *accurate* LC extrapolation to prevent premature stopping during the early stages of BO. To address this, we propose the use of **transfer learning** to maximize the sample efficiency of LC extrapolators.

As discussed in §2, among various plausible options, we employ PFNs [39] for LC extrapolation. Regarding the network architecture and training objective, we primarily follow ifBO [44], with more details deferred to Appendix E. A significant challenge in using PFNs for our purpose is that PFNs require relatively large Transformer architectures and massive amounts of training examples for strong generalization [2], making it *risky* to train them on a limited data.

**LC Mixup.** To overcome these challenges, we propose a novel transfer learning approach using the mixup strategy [66]. Assume that we have $M$ different training LC datasets and the corresponding $M$ sets of LCs collected from $N$ hyperparameter configurations. Let $l_n^{(m)} = [y_{n,1}^{(m)}, \ldots, y_{n,T}^{(m)}]$ be a $T$-dimensional vector representing the validation performances of the $m$-th dataset and the $n$-th configuration, forming a complete LC of length $T$. Define the matrix $L^{(m)} = [l_1^{(m)}; \ldots; l_N^{(m)}]^\top \in \mathbb{R}^{N \times T}$ as the stack of these LCs. To augment LCs, we propose two consecutive mixup strategies across datasets and configurations:

- **Dataset:** Sample a new LC dataset as $L' = \lambda_1 L^{(m)} + (1 - \lambda_1) L^{(m')}$, where $\lambda_1 \sim \text{Uniform}(0, 1)$ and $m, m' \in [M]$.
- **Configuration:** Sample a new configuration $x' = \lambda_2 x_n + (1 - \lambda_2) x_{n'}$ and its corresponding LC $l' = \lambda_2 l_n + (1 - \lambda_2) l_{n'}$, where $l_n$ and $l_{n'}$ denote the $n$-th and $n'$-th rows of $L'$, respectively, $\lambda_2 \sim \text{Uniform}(0, 1)$, and $n, n' \in [N]$.

Using this approach, we can sample infinitely many training examples $\{(x', l')\}$ by interpolating between LCs, resulting in a robust LC extrapolator with reduced overfitting.

# 4 Experiments

We next empirically validate the proposed method on various cost-sensitive multi-fidelity HPO settings. Our code is publicly available at `https://github.com/db-Lee/CFBO`.

## 4.1 Experimental Setups

**Datasets.** We evaluate CFBO on three standard LC benchmarks. **LCBench** [67]: contains learning curves of MLPs trained on multiple tabular datasets, **TaskSet** [37]: provides diverse optimization tasks across domains; we focus on 30 NLP tasks (text

Table 1: **Dataset overview.**

| Dataset | $d_x$ | $|\mathcal{X}|$ | $T$ | Train | Test |
|---|---|---|---|---|---|
| **LCBench** [67] | 7 | 2,000 | 51 | 20 datasets | 15 datasets |
| **TaskSet** [37] | 8 | 1,000 | 50 | 21 tasks | 9 tasks |
| **PD1** [56] | 4 | 240 | 50 | 16 tasks | 7 tasks |

classification and language modeling), and **PD1** [56]: includes learning curves of modern neural architectures, such as Transformers, trained on large-scale datasets (CIFAR-10/100 [31], ImageNet [45], and bioinformatics corpora). We split each benchmark into disjoint training and test tasks for transfer learning of LC extrapolators $p_\theta$. Detailed dataset statistics are summarized in Tab. 1, and additional descriptions are provided in Appendix D.

**Baselines without transfer learning.** We compare CFBO with a wide range of multi-fidelity HPO methods. **Random Search** [8] sequentially samples configurations at random. We also evaluate two Hyperband [34] variants: **BOHB** [14], which replaces random sampling with BO, and **DEHB** [4], which integrates evolutionary strategies for knowledge transfer. Among recent multi-fidelity BO methods, we include **DyHPO** [62], which combines a deep kernel Gaussian Process [GP; 58] with greedy short-horizon LC extrapolation; **DPL** [24], which fits power-law functions with ensemble modeling; and **ifBO** [44], a PFN-based [39] freeze-thaw BO method using PI acquisition at random future epochs. For a *fair comparison*, we use a **non-transfer (NT)** variant, **CFBO-NT**, with the LC extrapolator of ifBO: **excludes transfer-learning with LC mixup** in §3.4, but **retains $A(\cdot; U)$** (*cf.* Eq. 4 in §3.2) and **stopping criterion with the adaptive threshold $\delta_b$** (*cf.* Eqs. 5–7 in §3.3).

**Baselines with transfer learning.** Quick-Tune[†] [3] corresponds to the transfer-learning version of DyHPO, trained on the same LC datasets as our extrapolator. **FSBO** [60] is a black-box transfer-BO method that uses the same LC datasets to train a deep kernel GP surrogate. The both do not use the LC mixup in §3.4. The key difference from Quick-Tune[†] lies in the prediction target: FSBO predicts the final-epoch performance, whereas Quick-Tune[†] extrapolates the next-epoch performance.

**Implementation details** for CFBO and the baselines are provided in Appendix F.

**Utility function.** It is possible to manually collect user preference data and estimate the corresponding utility function (§3.1). In our experiments, however, we simplify this process by using a linear, quadratic or square root function, *i.e.*, $U(b, \tilde{y}_b) = \tilde{y}_b - \alpha \left(\frac{b}{B}\right)^c$, where $c \in \{1, 2, 0.5\}$, with $\alpha \in \{0, 2^{-6}, 2^{-5}, 2^{-4}, 2^{-3}, 2^{-2}\}$. Notably, setting $\alpha = 0$ removes any penalty associated with the budget $b$ spent during HPO, causing the HPO process to continue until the final step $B$, as in the conventional multi-fidelity HPO setup.

**Threshold hyperparameters $\delta$, $\gamma$, and $\beta$.** For the baselines, we set the threshold $\delta = 0.2$ in Eq. 5, as it performs well on the training split. For CFBO, we also use $\gamma = \log_{0.5} 0.2$, which corresponds to the adaptive threshold $\delta_b = 0.2$ when $p_b = 0.5$, to ensure a *fair comparison* with the baselines. We use $\beta = \exp(-1)$ for all experiments in this paper, except for the ablation study in Fig. 8d.

**Evaluation protocol.** We set the maximum budget[4] as $B = 300$ and report:

$$R := \frac{U_{\max} - U_p}{U_{\max} - U_{\min}} \in [0, 1], \text{ where } U_{\max} := \max_{n,t} U(t, y_{n,t}) \text{ and } U_{\min} \approx \min_n U(B, y_{n,1}), \quad (8)$$

which is similar to Eq. 5, but uses the *true* $U_{\max}$ and $U_{\min}$. Here, $U_p$ is the utility at termination (*e.g.*, after **break** in line 12 of Alg. 1). $U_{\max} := \max_{n,t} U(t, y_{n,t})$ denotes the maximum utility achievable by a single optimal configuration. Since exact computation of $U_{\min}$ is intractable, we approximate it as $\min_n U(B, y_{n,1})$, where $y_{n,1}$ is the first epoch performance. We report mean±std of normalized regrets over 5 runs (30 for Random, BOHB, and DEHB) and the average rank across all tasks.

---

[†]Modified version without pretrained model selection or wall-time balancing.

[4]While prior work [62, 44] uses $B = 1,000$, we set $B = 300$ to better show the benefit of transfer learning.

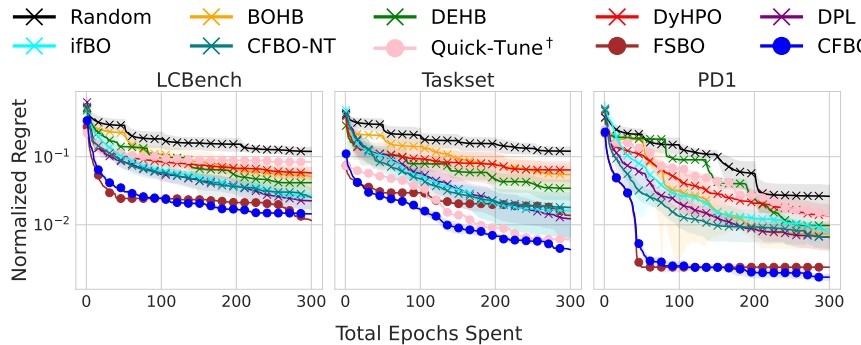

Figure 5: **Results on conventional multi-fidelity HPO** ($\alpha = 0$). **X** and **O** markers denote **non-transfer** and **transfer** learning methods, respectively.

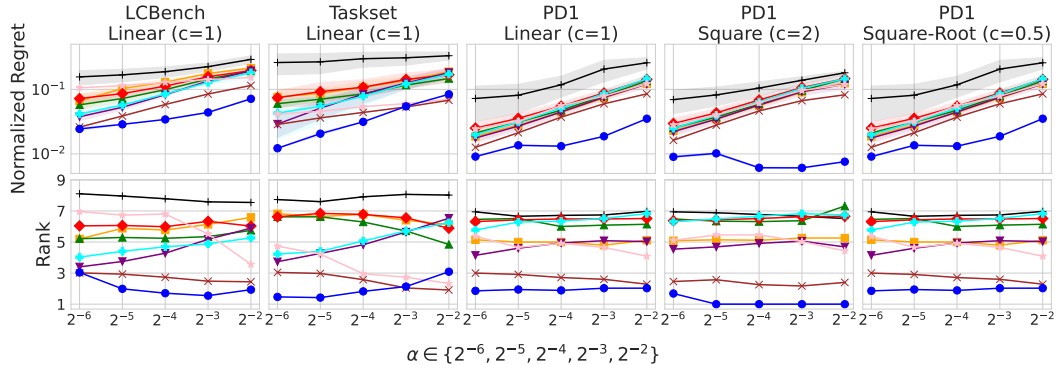

Figure 6: **Results on cost-sensitive multi-fidelity HPO** ($c \in \{1, 2, 0.5\}$ and $\alpha \in \{2^{-6}, 2^{-5}, 2^{-4}, 2^{-3}, 2^{-2}\}$).

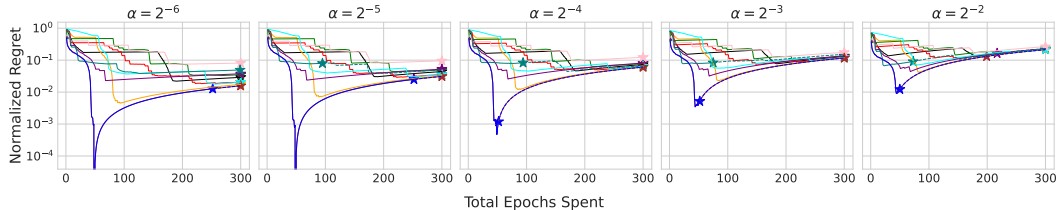

Figure 7: **Visualization of the normalized regrets on PD1 benchmark**. See Appendix G for the others.

## 4.2 Main Results

**Results on conventional multi-fidelity HPO ($\alpha = 0$).** We first validate the effectiveness of the proposed CFBO in conventional multi-fidelity HPO setups ($\alpha = 0$). The results are presented in Fig. 5, where **non-transfer** learning and **transfer** learning methods are denoted by **X** and **O** markers, respectively. Notably, although our methods are not specifically designed for conventional setups ($\alpha = 0$), they achieve results **comparable** to the baselines. For example, CFBO-NT, the non-transfer learning variant, performs on par with the best non–transfer baseline, ifBO. Similarly, CFBO achieves comparable performance to transfer-BO baselines, showing better normalized regret on TaskSet and PD1, and slightly higher regret on LCBench compared to FSBO.

**Results on cost-sensitive multi-fidelity HPO ($\alpha > 0$).** In real-world scenarios, utility functions can take various forms and may be defined differently by different users. To evaluate the effectiveness of CFBO under such realistic cost-sensitive settings, we conduct experiments on *various* utility functions, including linear ($c = 1$), square ($c = 2$), and square-root ($c = 0.5$), with $\alpha \in \{2^{-6}, 2^{-5}, 2^{-4}, 2^{-3}, 2^{-2}\}$. Fig. 6 presents the normalized regret (first row) and rank (second row) for each method. First, CFBO-NT not only **outperforms all non–transfer learning baselines** (denoted with the X markers) but also **surpasses a strong transfer-learning baseline** (*i.e.*, FSBO) in several settings—*e.g.*, most PD1 cases in terms of rank—except for $c = 2$ and $\alpha = 2^{-6}$. CFBO **exceeds all relevant baselines** in most settings, except TaskSet with $c = 1$ and $\alpha = 2^{-2}$, demonstrating the robustness of our method to variations in the utility function.

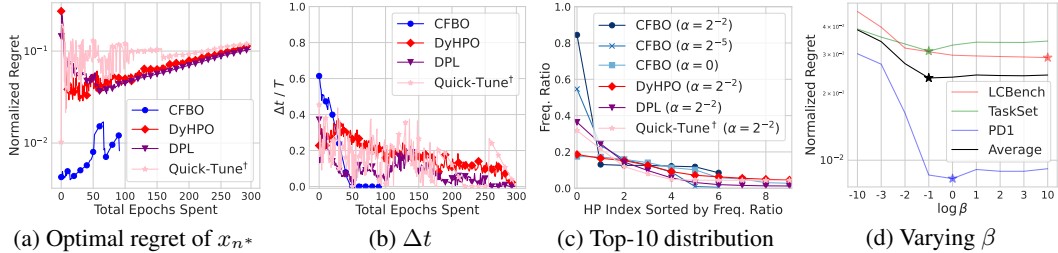

(a) Optimal regret of $x_{n*}$      (b) $\Delta t$      (c) Top-10 distribution      (d) Varying $\beta$

Figure 8: **(a) Optimal regret** of the selected configuration $x_{n*}$ at a future step $b + \Delta t$. **(b) Future $\Delta t$** in **(a)** at which the optimal regret of $x_{n*}$ is achieved. **(c) Distribution of the top-10** hyperparameter configurations selected during BO. **(d)** Normalized regrets with **varying $\beta$** of BetaCDF$(\cdot, \beta, \beta)^{\gamma}$ in Eq. 7 ($\gamma = \log_{0.5} 0.2$).

**Visualization.** We also visualize the normalized regret during the BO process using a task from the PD1 benchmark in Fig. 7. Asterisks indicate stopping points, while dotted lines represent the normalized regret achievable without stopping. Although our CFBO struggles to stop near the maximum utility (minimum regret) during the BO process when the penalty is weak ($\alpha \in \{2^{-6}, 2^{-5}\}$), the regrets obtained at those points are **still lower than the baselines**. In addition, CFBO successfully **stops almost at the optimum** when the penalty is stronger ($\alpha \in \{2^{-4}, 2^{-3}, 2^{-2}\}$).

**Algorithm runtime.** In Tab. 2, we report the average wall-clock time per BO step over five runs for DPL [24], ifBO [44], and CFBO. All measurements are conducted on a single A100 GPU. While ifBO is the most efficient method, the difference between the wall-clock times of ifBO and CFBO is **negligible**, as neural network training dominates the total wall-clock time in HPO (*e.g.*, 90 seconds for ResNet-50 [20] per training epoch in CIFAR-10/100 [31]).

Table 2: **Wall-clock time (seconds)** per BO step on LCBench, TaskSet, and PD1.

| Method | LCBench | TaskSet | PD1 |
|---|---|---|---|
| DPL [24] | $0.65_{\pm 0.02}$ | $0.64_{\pm 0.01}$ | $0.63_{\pm 0.01}$ |
| ifBO [44] | $0.58_{\pm 0.01}$ | $0.30_{\pm 0.00}$ | $0.08_{\pm 0.00}$ |
| **CFBO (ours)** | $1.52_{\pm 0.02}$ | $0.78_{\pm 0.01}$ | $0.23_{\pm 0.01}$ |

**Real-world object detection dataset.** We evaluate CFBO on a real-world object detection dataset and observe that it achieves the **best performance** (in terms of both normalized regret and rank) among all baselines. Details and additional experimental results are provided in Appendix G.

### 4.3 Analysis

**Ablation study.** To evaluate the effectiveness of each component, we conduct ablation studies on the proposed (1) **stopping criterion** (§3.3), (2) **acquisition function** (§3.2), and (3) **transfer learning with the LC mixup** (§3.4) on the PD1 benchmark. For (1) the stopping criterion, we compare two approaches: the smoothly mixed criterion with our adaptive threshold ($\delta_b$ ✓) and the only regret-based criterion with a fixed threshold $\delta$ ($\delta_b$ ✗), which is used by the baselines. For (2) the acquisition function, we use either our proposed approach (Eq. 4; $A$ ✓) or the acquisition function of ifBO [44, $A$ ✗]. For (3) transfer learning, we compare our surrogate trained with the proposed mixup strategy (T. ✓) against the ifBO [44] which is only trained on synthetic prior data (T. ✗).

In Tab. 3, comparing the first and second rows, we see that transfer learning is relatively more effective in conventional settings ($\alpha = 0$) than in cost-sensitive settings ($\alpha > 0$). Comparing between the second and third rows shows that our acquisition function enhances performance more in cost-sensitive settings. Lastly, comparing between the third and last row shows that our adaptive threshold significantly improves performance in cost-sensitive settings.

Table 3: **Results of ablation studies** using the PD1 benchmark. For better readability, we multiply 100 to normalized regrets.

| $\delta_b$ | $A$ | T. | $\alpha$ | | | | | |
|---|---|---|---|---|---|---|---|---|
| | | | 0 | $2^{-6}$ | $2^{-5}$ | $2^{-4}$ | $2^{-3}$ | $2^{-2}$ |
| ✗ | ✗ | ✗ | $0.8_{\pm 0.1}$ | $2.3_{\pm 0.1}$ | $3.7_{\pm 0.3}$ | $6.0_{\pm 0.6}$ | $9.8_{\pm 1.1}$ | $15.2_{\pm 2.0}$ |
| ✗ | ✗ | ✓ | $0.2_{\pm 0.0}$ | $1.7_{\pm 0.1}$ | $3.2_{\pm 0.1}$ | $5.9_{\pm 0.3}$ | $9.4_{\pm 0.4}$ | $11.7_{\pm 0.4}$ |
| ✗ | ✓ | ✓ | $0.2_{\pm 0.0}$ | $1.5_{\pm 0.0}$ | $2.6_{\pm 0.0}$ | $4.5_{\pm 0.0}$ | $6.9_{\pm 0.0}$ | $8.5_{\pm 0.0}$ |
| ✓ | ✓ | ✓ | $0.2_{\pm 0.0}$ | $\mathbf{1.0}_{\pm 0.0}$ | $\mathbf{1.3}_{\pm 0.0}$ | $\mathbf{0.9}_{\pm 0.0}$ | $\mathbf{1.1}_{\pm 0.0}$ | $\mathbf{1.7}_{\pm 0.0}$ |

**Analysis on acquisition function (§3.2).** To better understand the sources of improvement, we analyze the configurations selected by each method. Specifically, for each BO step $b$, we run the configuration $x_{n*}$ currently chosen at step $b$ up to its final epoch $T$ and compute two metrics: *the minimum ground-truth regret $R$ achievable at some future step $b + \Delta t$* (Fig. 8a) and the *corresponding optimal increment $\Delta t$* (Fig. 8b). In Fig. 8a, our method achieves significantly **lower minimum regret** compared to baselines. This indicates that our acquisition function in Eq. 4 performs *as intended*, selecting at each BO step the configuration expected to **maximally improve utility** in the

future. Fig. 8b shows that the configurations selected by our method initially correspond to larger $\Delta t$ values (*i.e.*, *non-greedy* behavior), but progressively shift to smaller $\Delta t$ values (*i.e.*, *greedy* behavior). This transition occurs because, as the BO progresses, the performance improvements begin to saturate, causing the cost of BO to dominate the trade-off. Consequently, $\Delta t$ becomes smaller, eventually approaching zero. These behaviors are **not prominently observed** in the baselines.

Fig. 8c shows the distribution of the top-10 most frequently selected configurations during BO. As expected, our method tends to **focus on a smaller subset** of configurations, optimizing for short-term performance, particularly under stronger penalties (*i.e.*, larger $\alpha$). The baselines exhibit excessive exploration of configurations, even under the strongest penalty (*i.e.*, $\alpha = 2^{-2}$).

**Analysis on stopping criterion (§3.3).** Next, we analyze the effectiveness of our stopping criterion with the adaptive threshold $\delta_b$ defined in Eqs. 5–7. Fig. 8d presents the normalized regret in various values of $\beta$, the mixing coefficient between the two extreme stopping criteria discussed in §3.3. $\log \beta \to 10$ corresponds to the baseline criterion (the fixed threshold $\delta = 0.2$), which relies solely on the estimated normalized regret in Eq. 5. $\log \beta \to -10$ corresponds to hard thresholding based exclusively on the PI $p_b$ in Eq. 7. The results show that the optimal criterion is achieved with **a smooth balance** between the two ($\beta = e^{-1}$), demonstrating the importance of incorporating the possibility of further improvement through $p_b$.

To further assess the significance of the stopping criterion in cost-sensitive multi-fidelity HPO, we quantify the improvement of normalized regrets obtained by an *optimal stopping criterion* for each method. The optimal stopping criterion assumes that the BO process is stopped at the point of *maximum utility* in the BO trajectory for each method. Fig. 9 presents the average improvement on the

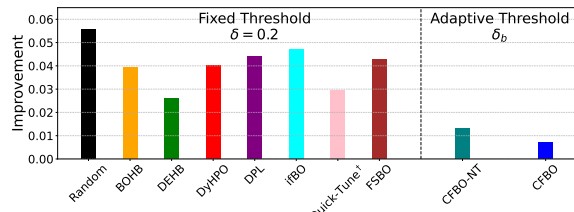

Figure 9: Improvement with **the optimal stopping criterion.**

PD1 benchmark across all $\alpha \in \{2^{-6}, 2^{-5}, 2^{-4}, 2^{-3}, 2^{-2}\}$. The improvement is smaller when using the adaptive threshold $\delta_b$ in Eq. 6; *i.e.*, our adaptive stopping criterion **closely approximates** the optimal stopping point, leaving minimal room for improvement and demonstrating its effectiveness.

## 5 Conclusion

In this paper, we present cost-sensitive freeze-thaw Bayesian optimization to improve the efficiency of hyperparameter optimization (HPO). Assuming that users aim to early-stop HPO when the utility saturates, we introduced a novel acquisition function and stopping criterion specifically tailored to this problem setup. Additionally, we proposed a novel transfer learning method for training a sample-efficient in-context learning curve (LC) extrapolator. Our empirical evaluation demonstrated the effectiveness of our approach compared to existing multi-fidelity HPO and transfer-BO methods.

**Limitations and future work.** We identify three main limitations of our study and outline potential directions for future research as follows:

- We focus on **pool-based HPO**, *i.e.*, $|\mathcal{X}| \in \mathbb{N}$, which may not fully reflect real-world applications. To enable continuous optimization over $x_n \in \mathbb{R}^{d_x}$, a common approach [48] is to perform *gradient ascent* on the hyperparameter configuration $x_n$ to maximize the acquisition function $A$. In this case, $A$ must be differentiable with respect to $x_n$. Unfortunately, our acquisition function $A(\cdot; U)$ in Eq. 4 does *not satisfy* this property, since the sampling from $p_\theta$ (for MC estimation) is not differentiable. Extending CFBO to this continuous setting remains an interesting direction.

- In §3.4, we explore transfer learning by training Prior-Data Fitted Networks [39] on mixed LC datasets. Recently, Tune-Tables [15] have demonstrated strong performance in fine-tuning PFNs on tabular datasets through **context optimization**. Since transfer learning is not only a key component of our work but also widely recognized in the literature [60, 3], integrating the LC mixup with the context optimization approach of Tune-Tables presents a promising direction.

- Our work assumes that users care only about the trade-off within a single task. In real-world scenarios, however, users (*e.g.*, cloud service customers) may also consider **future budgets** or the costs of **multiple tasks** running in parallel. Developing a system that accounts for these more complex and realistic settings would be a valuable direction.

## Acknowledgement

We express our sincere gratitude to the anonymous reviewers (**uqhd**, **AJQp**, **Ly5J**, and **Vxb6**) for their valuable feedback and efforts in helping us improve this paper.

**Funding.** This work was supported by Institute for Information & communications Technology Planning & Evaluation(IITP) grant funded by the Korea government(MSIT) (RS-2019-II190075, Artificial Intelligence Graduate School Program(KAIST), No.RS-2022-II220713, Meta-learning Applicable to Real-world Problems), IITP with a grant funded by the Ministry of Science and ICT (MSIT) of the Republic of Korea in connection with the Global AI Frontier Lab International Collaborative Research (No. RS-2024-00469482 & RS-2024-00509279), National Research Foundation of Korea (NRF) grant funded by MSIT (No. RS-2023-00256259), and Center for Applied Research in Artificial Intelligence (CARAI) grant funded by DAPA and ADD (UD190031RD).

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

## Appendix Overview

This appendix provides supplementary materials to support the main paper as follows:

- **Related Work** (Appendix A): discusses the relevant literature on (1) multi-fidelity HPO, (2) freeze-thaw BO, (3) LC extrapolation, (4) transfer BO, (5) cost-sensitive HPO, (6) early stopping BO, (7) BO with user preferences, and (8) neural processes.
- **Notation** (Appendix B): summarizes an overview of the notation used throughout this paper.
- **Utility Estimation** (Appendix C): details how we can estimate user utility with Bradley-Terry model [10] and provides several examples.
- **Dataset** (Appendix D): provides details on benchmarks (LCBench [67], TaskSet [37], PD1 [55]) and data preprocessing.
- **Details on LC Extrapolator** (Appendix E): contains the architectural and training details of the LC extrapolator $p_\theta$.
- **Implementation Details** (Appendix F): includes implementation details of CFBO and baselines.
- **Additional Experiments** (Appendix G): presents additional experiments, including results on real-world object detection datasets and visualizations of normalized regrets and LC extrapolations during Bayesian optimization process.

## A  Related Work

**Multi-fidelity HPO.**   Unlike traditional black-box approaches for HPO [8, 22, 9, 48, 50, 49, 12, 40], multi-fidelity (or gray-box) HPO aims to optimize hyperparameters in a sample-efficient manner by utilizing low-fidelity information (*e.g.*, validation set performances with smaller training datasets) as a proxy for higher or full fidelities [52, 25, 29, 43, 26, 63, 64], dramatically speeding up the HPO process. In this paper, we focus on using performances at fewer training epochs to better predict and optimize performances at longer training epochs. One well-known example is Hyperband [34], a bandit-based method that randomly selects a set of hyperparameter configurations and stops poorly performing ones using successive halving [27] before reaching the last training epoch. While Hyperband shows much better performance than random search [8], its computational or sample efficiency can be further improved by replacing random sampling of configurations with Bayesian optimization [BOHB; 14], adopting an evolution strategy to promote internal knowledge transfer [DEHB; 4], or making it asynchronously parallel [33].

**Freeze-thaw BO.**   Freeze-thaw BO [53] dynamically pauses (freezes) and resumes (thaws) configurations based on the last epoch's performances extrapolated from a set of partially observed learning curves (LCs) obtained from other configurations, leading to an efficient and sensible allocation of computational resources. DyHPO [62] and its transfer version [3] improve the computational efficiency of freeze-thaw BO using deep kernel Gaussian processes [GPs; 58], but their acquisition maximizes one-step forward fidelity (*i.e.*, epoch), producing a myopic strategy. Other recent variants of freeze-thaw BO include DPL [24] and ifBO [44], which are non-myopic, and their acquisitions maximize performance either at the last fidelity or random future steps. On the other hand, we maximize the utility specified by each user.

**LC extrapolation.**   Freeze-thaw BO requires the ability to dynamically update predictions on future performances from partially observed learning curves (LCs), thus heavily relying on LC extrapolation [7, 17, 61]. DyHPO [62] and Quick-Tune [3] propose to extrapolate LCs for only a single step forward. Freeze-thaw BO [53] and DPL [24] use non-greedy extrapolations but limit the shape of the LCs. [13] consider a broader set of basis functions but require computationally expensive Markov Chain Monte Carlo (MCMC), and also do not consider correlations between different configurations. [30] models interactions between configurations with Bayesian neural networks (BNNs) but suffers from the same computational inefficiency of MCMC and online retraining. LC-PFNs [2] are an in-context Bayesian LC extrapolation method without retraining, but they do not consider interactions between configurations. Recently, ifBO [44] further combined LC-PFNs with PFNs4BO [40] to develop an in-context surrogate function for freeze-thaw BO, but they train PFNs only with a prior distribution. On the other hand, we use transfer learning with LC mixup (§3.4), *i.e.*, training PFNs with existing LC datasets, to improve the sample efficiency of freeze-thaw BO while successfully encoding the correlations between configurations at the same time.

**Transfer-BO.** Transfer learning can be used to improve the sample efficiency of BO [5], and here we list a few examples. Some recent work has explored scalable transfer learning with deep neural networks [42, 60]. Additionally, different components of BO can be transferred, such as observations [52], surrogate functions [19, 60], hyperparameter initializations [60], or all of them [57]. However, most of the existing transfer-BO approaches assume traditional black-box BO settings. To the best of our knowledge, Quick-Tune [3] is the only recent work that targets multi-fidelity and transfer-BO simultaneously. However, their multi-fidelity BO formulation is greedy. As described in Figs. 8a and 8b of §4.3, our CFBO can dynamically control the degree of greediness during BO by explicitly taking into account the trade-off between the cost and performance of BO.

**Cost-sensitive HPO.** Multi-fidelity BO is inherently cost-sensitive since predictions get more accurate as the gap between fidelities becomes smaller. However, the performance metric of such vanilla multi-fidelity BO monotonically increases as we spend more budget. In this paper, we aim to find the *optimal trade-off* between the budget spent and the corresponding intermediate performances of BO, thereby automatically early-stopping the BO around the maximal utility. Quick-Tune [3] also suggests cost-sensitive BO in multi-fidelity settings, but unlike our work, their primary focus is on the trade-off between performance and the cost of BO associated with *pretrained models of various sizes*, which can be seen as a generalization of the traditional notion of cost-sensitive BO [48, 1, 32, 65] from black-box to multi-fidelity settings.

**Early stopping BO.** Makarova et al. [35] propose a *principled* early stopping BO using:

$$f^* - f_b \leq 2\epsilon + \tilde{y}^* - \tilde{y}_b, \tag{9}$$

where $f$ denotes the true performance of population, $\tilde{y}_b$ the validation performance, $*$ the maximizer, and $\epsilon \in \mathbb{R}_{>0}$ a statistical error term. Eq. 9 leads to stopping condition as follows:

$$\tilde{y}^* - \tilde{y}_b \leq \epsilon. \tag{10}$$

Eq. 10 can be expressed using a utility function:

$$U(b, \tilde{y}_b) = \begin{cases} \tilde{y}_b, & \text{if } \tilde{y}_b \geq \tilde{y}^* - \epsilon, \\ -\infty, & \text{otherwise.} \end{cases} \tag{11}$$

Since $\tilde{y}^*$ and $\epsilon$ are unknown, Makarova et al. [35] estimate them using lower confidence bound (LCB), upper confidence bound (UCB), and the coefficient of variation (CV). As these estimators are functions of the budget $b$, Eq. 11 can be rewritten in a **utility view**:

$$U(b, \tilde{y}_b) = \begin{cases} \tilde{y}_b, & \text{if } \tilde{y}_b \geq c(b), \\ -\infty, & \text{otherwise.} \end{cases} \tag{12}$$

Interpreted this way, the stopping criterion in Makarova et al. [35] *roughly* corresponds to the preference: *"Stop once performance exceeds $c(b)$; additional budget has no value thereafter."* Therefore, Eq. 12 represents a special case of our utility formulation.

Similarly, Wilson [59] roughly fits into a utility view:

$$U(b, \tilde{y}_b) = \begin{cases} \tilde{y}_b, & \text{if } p(r_b \leq \epsilon | \mathcal{C}) < 1 - \delta, \\ -c_b, & \text{otherwise,} \end{cases} \tag{13}$$

where $r_b = f^* - f_b$ and $c_b$ is the cumulative cost. Furthermore, although both Makarova et al. [35] and Wilson [59] provide *principled and general* approaches for early stopping in BO, they primarily target *black-box* BO settings. Due to this fundamental difference, we exclude them from our baselines in §4.

**BO with user preference.** Several works have tried to encode the user beliefs about hyperparameter configurations into BO frameworks [51, 23, 36]. On the other hand, our paper suggests encoding user preferences regarding the trade-off between cost and performance. Therefore, the notion of user preference in this paper is largely different from the previous literature.

**Neural Process** Our training method is more similar to Transformer Neural Processes (TNPs) [41], a Transformer variant of Neural Processes [NPs; 18]. Similarly to PFNs [39, 2, 44], TNPs directly maximize the likelihood of target data given context data with a Transformer architecture, which differs from the typical NP variants that summarize the context into a latent variable and perform variational inference on it. Moreover, as with the other NP variants, TNPs meta-learn a model over a distribution of tasks to perform sample-efficient inference. In this vein, the whole training pipeline of our LC extrapolator can be seen as an instance of TNPs—we also meta-learn a sample-efficient LC extrapolator over the distribution of LCs induced by the mixup strategy.

# B  Notation

In this section, we summarize the notation used throughout the paper in Tab. 4.

Table 4: **Notation summary**.

| Notation | Description |
|---|---|
| $x_n \in \mathbb{R}^{d_x}$ | $n$-th hyperparameter configuration with dimension of $d_x \in \mathbb{N}$ |
| $\mathcal{X} = \{x_1, \dots, x_N\}$ | Set of hyperparameter configurations |
| $T \in \mathbb{N}$ | Maximum training epochs for each configuration |
| $t \in [T] := \{1, \dots, T\}$ | Training-epoch index |
| $y_{n,t} \in [0, 1]$ | Validation performance of $x_n$ at epoch $t$ |
| $B \in \mathbb{N}$ | Maximum total training epochs (overall BO budget) |
| $b \in [B]$ | Budget spent so far |
| $\tilde{y}_b$ | (Cumulative) best performance observed up to budget $b$ |
| $\mathcal{C} = \{(x, t, y)\}$ | Collected partial or full learning curves |
| $p_\theta : \mathbb{R}^{d_x} \times [T] \times \mathcal{C} \to [0, 1]$ | Learning-curve extrapolator |
| $U : [B] \times [0, 1] \to [0, 1]$ | Utility function |
| $c \in \mathbb{R}$ | Hyperparameter which decides for functional form of $U(b, \tilde{y}_b) = \tilde{y}_b - \alpha(\frac{b}{B})^c$ |
| $\alpha \in [0, 1]$ | Penalty coefficient of $U(b, \tilde{y}_b) = \tilde{y}_b - \alpha(\frac{b}{B})^c$ |
| $U_p \in [0, 1]$ | Utility immediately before current BO step |
| $A(\cdot; U) : [N] \to \mathbb{R}$ | Acquisition function in Eq. 4 |
| $\hat{R}_b := \frac{\hat{U}_{\max} - U_p}{\hat{U}_{\max} - \hat{U}_{\min}} \in [0, 1]$ | The roughly estimated normalized regret in Eq. 5 |
| $\hat{U}_{\max} \in [0, 1]$ | Maximum utility observed so far for computing $\hat{R}_b$ in Eq. 5 |
| $\hat{U}_{\min} := U(B, \tilde{y}_1) \in [0, 1]$ | Approximated minimum utility for computing $\hat{R}_b$ in Eq. 5 |
| $R := \frac{U_{\max} - U_p}{\hat{U}_{\max} - U_{\min}} \in [0, 1]$ | the true normalized regret in Eq. 5 |
| $U_{\max} := \max_{n,t} U(t, y_{n,t}) \in [0, 1]$ | True maximum utility for computing $R$ in Eq. 8 |
| $U_{\min} \approx \min_n U(B, y_{n,1}) \in [0, 1]$ | Approximated true minimum utility for computing $R$ in Eq. 8 |
| $\delta \in [0, 1]$ | Fixed stopping threshold for baselines in Eq. 5 |
| $\delta_b := \mathrm{BetaCDF}(p_b, \beta, \beta)^\gamma \in [0, 1]$ | Adaptive stopping threshold instead of $\delta$ in Eq. 5 |
| $p_b \in [0, 1]$ | Probability that current configuration will improve $U_p$ in Eq. 7 |
| $\mathrm{BetaCDF}(\cdot; \beta, \beta) : [0, 1] \to [0, 1]$ | the CDF of Beta distribution with shape $\beta$ |
| $\beta \in \mathbb{R}_{>0}$ | Hyperparameter controlling interpolation in Eq. 6 |
| $\gamma \in \mathbb{R}_{>0}$ | Hyperparameter setting $\delta_b$ at $p_b = 0.5$ in Eq. 6 |

## C  Utility Estimation

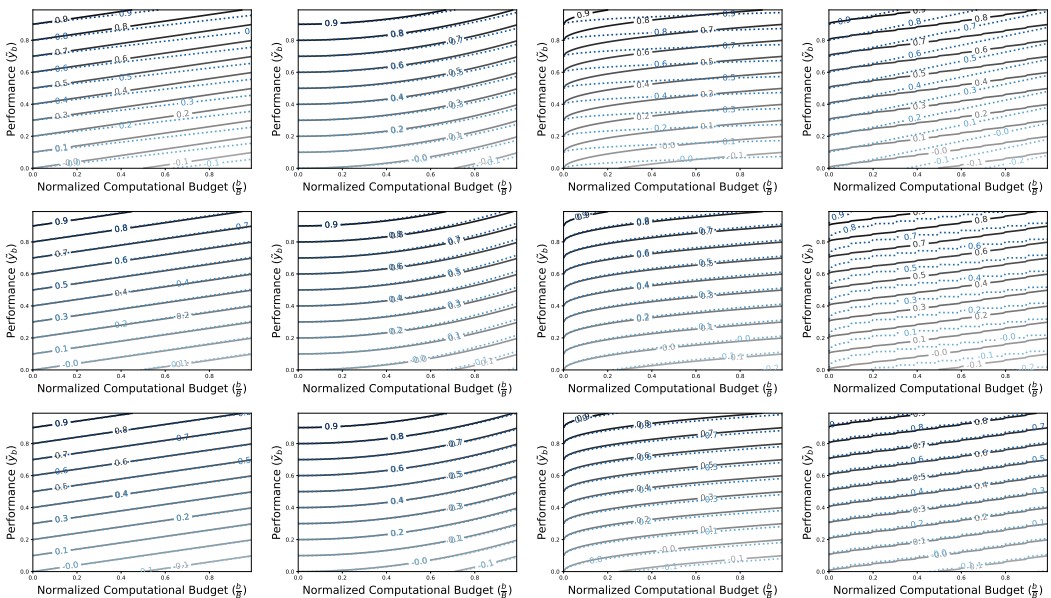

Figure 10: **Contour plots of true utilities and their approximations**. From left to right, the columns show **different functional forms** of linear ($c = 1$), quadratic ($c = 2$), square root ($c = 0.5$), and a combination of four different functions including a staircase function. From top to bottom, the rows represent **30**, **100**, and **1000** user preference data pairs.

**Functional forms.** In real-world scenarios, users can have more complex utility functions. We therefore consider the following functional forms: **(1) Linear**: $\tilde{y}_b - \alpha\frac{b}{B}$, **(2) Quadratic**: $\tilde{y}_b - \alpha(\frac{b}{B})^2$, **(3) Square-Root**: $\tilde{y}_b - \alpha(\frac{b}{B})^{0.5}$, and **(4) Staircase**: $\tilde{y}_b - \sum_i \alpha_i \mathbb{1}(b \in A_i)$ ($\mathbb{1}(\cdot)$ is an indicator function, and $A_i$ is an $i$-th interval). We further assume that these utility functions can be linearly combined, *e.g.*, $U(\cdot, \cdot) = w^{(\text{linear})} U^{(\text{linear})}(\cdot, \cdot) + \ldots + w^{(\text{staircase})} U^{(\text{staircase})}(\cdot, \cdot)$, where $w^{(\text{linear})} + \ldots + w^{(\text{staircase})} = 1$.

**Data collection.** We now describe how we roughly estimate user utility based on the user preference data pairs. First of all, we assume that it is possible for users to decide whether they prefer one point to the other one, instead of quantifying their utility, *i.e.*, we can collect user preference data. For simulation, we assume that we are given these preference data generated by true utility function. True utility function $U$ is randomly defined by sampling penalizing coefficient from $\text{Uniform}(0, 1)$ and linear combination coefficients from Dirichlet distribution. We randomly select *meaningful* data pairs from $b/B \sim \text{Uniform}(0, 1)$ and $\tilde{y}_b \sim \text{Uniform}(0, 1)$. Here, "meaningful data pairs" means that one datapoint of each pair is not trivially preferred by user: for example, one has larger performance $\tilde{y}_b$ with smaller budget $b$ than the other. Users then label their preference on these data pairs; for simulation, we label the pairs by using the true utility functions with sampled $\alpha$s and linear combination coefficients $w$s.

**Training details.** As explained in Eqs. 2 and 3, we use the binary cross-entropy loss $\ell(x, y) \coloneqq -y \log x - (1 - y) \log(1 - x)$ in Eq. 3 between the probability of preference described by the BT model and the preference label $y_> \in \{0, 1\}$. We begin by randomly initializing another utility function to approximate a randomly sampled true utility function, setting the $w$s and $\alpha$s to $\frac{1}{4}$ and 0.0001, respectively. We use gradient-based optimization algorithm (*e.g.*, SGD) with 1000 iterations. The temperature term $\tau$ in Eq. 2 is set to 0.05.

**Estimation results.** Fig. 10 demonstrates that not only can single utilities—linear, quadratic, and square root—be well approximated using preference data, but even more complex utilities (*e.g.*, a combination of four different utilities) can also be accurately approximated. Furthermore, we found that the approximation works well even with smaller numbers (*e.g.*, 30, 100) of user preference data pairs for simpler cases (*i.e.*, single utilities).

# D   Dataset

**LCBench.**   We use the LCBench benchmark [67], which consists of learning curves for MLPs trained on multiple tabular datasets. Each task contains 2,000 learning curves with 51 training epochs. We summarize the hyperparameter settings of LCBench in Tab. 5.

Table 5: The $d_x = 7$ hyperparameters for **LCBench** datasets.

| Name | Type | Vaules | Info |
|---|---|---|---|
| batch_size | integer | $[2^4, 2^9]$ | log |
| learning_rate | continuous | $[10^{-4}, 10^{-1}]$ | log |
| max_dropout | continuous | $[0, 1]$ | |
| max_units | integer | $[2^6, 2^{10}]$ | log |
| momentum | continuous | $[0.1, 0.99]$ | |
| max_layers | integer | $[1, 5]$ | |
| weight_decay | continuous | $[10^{-5}, 10^{-1}]$ | |

The training/test splits are as follows:

- **Training datasets:**
  APSFailure, Amazon_employee_access, Australian, Fashion-MNIST,
  KDDCup09_appetency, MiniBooNE, adult, airlines, albert, bank-marketing,
  blood-transfusion-service-center, car, christine, cnae-9, connect-4,
  covertype, credit-g, dionis, fabert, helena.

- **Test datasets:**
  higgs, jannis, jasmine, jungle_chess_2pcs_raw_endgame_complete, kc1,
  kr-vs-kp, mfeat-factors, nomao, numerai28.6, phoneme, segment, shuttle,
  sylvine, vehicle, volkert.

**TaskSet.**   We use the TaskSet benchmark [37], which provides learning curves from diverse optimization tasks across multiple domains. Each task contains 1,000 learning curves with 50 training epochs. We summarize the hyperparameter settings of TaskSet in Tab. 6.

Table 6: The $d_x = 8$ hyperparameters for **TaskSet** tasks.

| Name | Type | Vaules | Info |
|---|---|---|---|
| learning_rate | continuous | $[10^{-9}, 10^1]$ | log |
| beta1 | continuous | $[10^{-4}, 1]$ | |
| beta2 | continuous | $[10^{-3}, 1]$ | |
| epsilon | continuous | $[10^{-12}, 10^3]$ | log |
| l1 | continuous | $[10^{-9}, 10^1]$ | log |
| l2 | continuous | $[10^{-9}, 10^1]$ | log |
| linear_decay | continuous | $[10^{-8}, 10^{-4}]$ | log |
| exponential_decay | continuous | $[10^{-6}, 10^{-3}]$ | log |

The training/test splits are as follows:

- **Training tasks:**
  rnn_text_classification_family_seed$\{$19, 3, 46, 47, 59, 6, 66$\}$,
  word_rnn_language_model_family_seed$\{$22, 47, 48, 74, 76, 81$\}$,
  char_rnn_language_model_family_seed$\{$19, 26, 31, 42, 48, 5, 74$\}$.

- **Test tasks:**
  rnn_text_classification_family_seed$\{$8, 82, 89$\}$,
  word_rnn_language_model_family_seed$\{$84, 98, 99$\}$,
  char_rnn_language_model_family_seed$\{$84, 94, 96$\}$.

**PD1.**   We use the PD1 benchmark [56], where each task contains 240 LCs with 50 training epochs. To facilitate transfer learning, we preprocess the LC data of PD1 by excluding hyperparameter configurations with their training diverging (*e.g.*, LCs with NaN), and linearly interpolate the LCs to match their length across different tasks. We then obtain the LCs of 50 epochs over the 240 configurations. We summarize the hyperparameter of PD1 in Table 7.

Table 7: The $d_x = 8$ hyperparameters for **PD1** tasks.

| Name | Type | Vaules | Info |
|---|---|---|---|
| lr_initial_value | continuous | $[10^{-5}, 10^1]$ | log |
| lr_power | continuous | $[10^{-1}, 2]$ | |
| lr_decay_steps_factor | continuous | $[10^{-2}, 0.99]$ | |
| one_minus_momentum | continuous | $[10^{-5}, 1]$ | log |

The training/test splits are as follows:

- **Training tasks:**
  uniref50_transformer_batch_size_128, lm1b_transformer_batch_size_2048,
  imagenet_resnet_batch_size_256, mnist_max_pooling_cnn_tanh_batch_size_2048,
  mnist_max_pooling_cnn_relu_batch_size_{256, 2048},
  mnist_simple_cnn_batch_size_{256, 2048},
  fashion_mnist_max_pooling_cnn_tanh_batch_size_2048,
  fashion_mnist_max_pooling_cnn_relu_batch_size_{256, 2048},
  fashion_mnist_simple_cnn_batch_size_{256, 2048},
  svhn_no_extra_wide_resnet_batch_size_1024,
  cifar{10,100}_wide_resnet_batch_size_2048.

- **Test tasks:**
  imagenet_resnet_batch_size_512, translate_wmt_xformer_translate_batch_size_64,
  mnist_max_pooling_cnn_tanh_batch_size_256,
  fashion_mnist_max_pooling_cnn_tanh_batch_size_256,
  svhn_no_extra_wide_resnet_batch_size_256,
  cifar100_wide_resnet_batch_size_256, cifar10_wide_resnet_batch_size_256.

**Data Preprocessing**   We follow the convention of LC-PFN [2] for data preprocessing; we consistently apply a non-linear LC normalization[5] to the LC data of three benchmarks, which not only maps either accuracy or log-loss LCs into $[0, 1]$ but also simply makes our optimization as a maximization problem. We also use the maximum and minimum values for each task in LCBench and PD1 benchmark for the LC normalization. In TaskSet, we only use the $y_{n,0}$ (*i.e.*, the initial log-loss without taking any gradient steps) for the LC normalization.

# E   Details on LC Extrapolator

**Construction of context and query points.**   Following ifBO [44], we can simulate each step of BO; predicting the remaining part of LC in all configurations conditioned on the set $\mathcal{C}$ of the collected partial LCs. To do so, we construct a training task by randomly sampling the context and query points from the LC benchmarks after the proposed LC mixup in §3.4 as follows:

1. We choose an LC dataset $L = [l_1, \ldots, l_N]^\top \in \mathbb{R}^{N \times T}$ by randomly sampling $m \in [M]$.
2. From $L^{(m)}$, we randomly sample $n_1, \ldots, n_C \in [N]$ and $t_1, ..., t_C \in [T]$ and construct context points of $X^{(c)} = [x_{n_1}, \ldots, x_{n_C}]^\top \in \mathbb{R}^{C \times d_x}$, $T^{(c)} = [t_1/T, \ldots, t_C/T]^\top \in \mathbb{R}^{C \times 1}$, and $Y^{(c)} = [y_{n_1,t_1}, \ldots, y_{n_C,t_C}] \in \mathbb{R}^{C \times 1}$.
3. From the chosen $L$, we exclude $n_1, \ldots, n_C \in [N]$ and $t_1, ..., t_C \in [T]$ and randomly sample $n'_1, \ldots, n'_Q \in [N]$ and $t'_1, ..., t'_Q \in [T]$ and construct query points of $X^{(q)} = [x_{n'_1}, \ldots, x_{n'_Q}]^\top \in \mathbb{R}^{Q \times d_x}$, $T^{(q)} = [t'_1/T, \ldots, t'_C/T]^\top \in \mathbb{R}^{Q \times 1}$, and $Y^{(q)} = [y_{n'_1,t'_1}, \ldots, y_{n'_Q,t'_Q}] \in \mathbb{R}^{Q \times 1}$.

**Architecture for predicting LCs.**   From now on, we denote any row vectors of the constructed context and query points in lowercase, *e.g.*, $x^{(q)}$ of $X^{(q)}$, or $y^{(q)}$ of $Y^{(q)}$. We train a Transformer [54] that is a probabilistic model of $f(Y^{(q)}|X^{(c)}, T^{(c)}, Y^{(c)}, X^{(q)}, T^{(q)})$. Conditioned on any subsets of LCs (*i.e.*, $X^{(c)}, T^{(c)}$, and $Y^{(c)}$), this model predicts a mini-batch of the remaining part of LCs of existing hyperparameter configurations in a given dataset (*i.e.*, $Y^{(q)}$ of $X^{(q)}$ and $T^{(q)}$). For

---

[5]The details can be found in Appendix A of PFN [2] and https://github.com/automl/lcpfn/blob/main/lcpfn/utils.py.

computational efficiency, we further assume that the query points are independent of each other, as done in PFNs [39, 44]:

$$p_\theta(Y^{(q)}|X^{(c)}, T^{(c)}, Y^{(c)}, X^{(q)}, T^{(q)}) = \prod_{x^{(q)}, t^{(q)}, y^{(q)}} p_\theta(y^{(q)}|x^{(q)}, t^{(q)}, X^{(c)}, T^{(c)}, Y^{(c)}). \quad (14)$$

Before encoding the input into the Transformer, we first encode the input of $X^{(c)}, T^{(c)}, Y^{(c)}, X^{(q)}$, and $T^{(q)}$ using a simple linear layer as follows:

$$H^{(c)} = X^{(c)}W_x + T^{(c)}W_t + Y^{(c)}W_y \quad (15)$$

$$H^{(q)} = X^{(q)}W_x + T^{(q)}W_t, \quad (16)$$

where $W_x \in \mathbb{R}^{d_x \times d_h}$, $W_t \in \mathbb{R}^{1 \times d_h}$, and $W_y \in \mathbb{R}^{1 \times d_h}$. Here, we abbreviate the bias term.

Then we concatenate the encoded representations of $H^{(c)}$ and $H^{(q)}$, and feedforward it into the Transformer layer by treating each pair of each row vector of $H^{(c)}$ and $H^{(q)}$ as a separate position/token as follows:

$$H = \texttt{Transformer}([H^{(c)}; H^{(q)}, \texttt{Mask}]) \in \mathbb{R}^{(M+N) \times d_h} \quad (17)$$

$$\hat{Y} = \texttt{Head}(H) \in \mathbb{R}^{(M+N) \times d_o}, \quad (18)$$

where $\texttt{Transformer}(\cdot)$ and $\texttt{Head}(\cdot)$ denote the Transformer layer and multi-layer perceptron (MLP) for the output prediction, respectively. $\texttt{Mask} \in \mathbb{R}^{(N_c+N_q) \times (N_c+N_q)}$ is the transformer mask that allows all tokens to attend context tokens only.

In Eq. 18, the output dimension $d_o$ is specified by the output distribution of $y \in [0, 1]$. Following PFNs [39, 2, 44], we discretize the domain of $y$ by $d_o = 1000$ and use the categorical distribution. Finally, we only take the output of the last $N_q$ tokens as the output, *i.e.*, $\hat{Y}^{(q)} = \hat{Y}[:, N_c : (N_c + N_q)] \in \mathbb{R}^{N_q \times d_h}$ (using the indexing operation), since we only need the output of query tokens for modeling $\prod p_\theta(y^{(q)}|x^{(q)}, t^{(q)}, X^{(c)}, T^{(c)}, Y^{(c)})$.

**Training Objective.** Our training objective is then defined as follows:

$$\arg\min_\theta \mathbb{E}_p \left[ -\sum_{x^{(q)}, t^{(q)}, y^{(q)}} \log p_\theta(y^{(q)}|x^{(q)}, t^{(q)}, X^{(c)}, T^{(c)}, Y^{(c)}) \right] + \lambda_{\text{PFN}} \mathcal{L}_{\text{PFN}}, \quad (19)$$

where $p$ is the empirical distribution of LC data with the LC mixup in §3.4. We additionally minimize $\mathcal{L}_{\text{PFN}}$ with coefficient $\lambda_{\text{PFN}}$, which is the LC extrapolation loss of LC-PFN [2] for each LC. We found $\lambda_{\text{PFN}} = 0.1$ works well for most cases.

**Training Details.** We sample 4 training tasks for each iteration, *i.e.*, the size of meta mini-batch is set to 4. We uniformly sample the size $C$ of context points from 1 to 300, and the size of query points $Q$ is set to 2,048. Following LC-PFN [2], the hidden size of each Transformer block $d_h$, the hidden size of feed-forward networks, and the number of layers of Transformer, dropout rate are set 1,024, 2,048, 12, and 0.2, respectively. We use GeLU activation [21]. We train the extrapolator for 100,000 iterations on training split of each benchmark with Adam [28] optimizer. The $\ell_2$ norm of the gradient is clipped to 1.0. The learning rate is linearly increased to $2 \cdot 10^{-05}$ for 25,000 iterations (25% of the total iteration), and it is decreased with a cosine scheduling until the end. The whole training process takes roughly 10 hours in a single A100 GPU.

# F  Implementation Details

**0-epoch LC value.** We assume access to the 0-epoch LC value $(y_{n,0})$ which is the performance without taking any gradient steps, *i.e.*, *random guessing*. This is also plausible for realistic scenarios, since in most deep-learning models *one evaluation cost* is acceptable in comparison to training costs. The 0-epoch LC values originally are not provided except for LCBench; we use the log-loss of the first epoch as the 0-epoch LC value for TaskSet, as it is already sufficiently large in our chosen tasks. For PD1, we interpolate the LCs to be the length of 51 training epochs, and we take the first performance as the 0-epoch LC value. Furthermore, we take the average over the 0-epoch LC values $\bar{y}_0 := \frac{1}{N} \sum_{n=1}^{N} y_{n,0}$ for convenience. The average 0-epoch LC value $\bar{y}_0$ is always conditioned on our LC extrapolator $p_\theta$ for both the training stage in Appendix E and the BO stage in Alg. 1.

**Monte-Carlo (MC) sampling for reducing variance of LCs.** We estimate the expectation of the proposed acquisition function $A(\cdot; U)$ in Eq. 4 with 1,000 MC samples. We found that each LC $y_{n,t_{n:T}}$ extrapolated from the LC extrapolator $p_\theta(\cdot|x_n, \mathcal{C})$ is *noisy*, due to the assumption that the query points of $y_{n,t_{n:T}}$ are independent of each other in Eq. 14. Since we compute $\tilde{y}_{b+\Delta t}$ by taking the maximum among the last step BO performance (*i.e.*, cumulative max operation), the quality of the estimation degenerates significantly due to the noise. To avoid this, we reduce the variance of the MC samples by taking the *average* of the sampled LCs. For example, we sample 5,000 LC samples from the LC extrapolator $p_\theta$, then divide them into 1,000 groups and take the average among the 5 LC samples in each group. We empirically found that this stabilize the estimation of not only acquisition function $A(\cdot; U)$ and probability of utility improvement $p_b$ in Eq. 7.

**Implementation details for baselines.** We list the implementation details for baselines as follows:

1. **Random Search** [48]: Instead of randomly selecting a hyperparameter configuration for each BO step, we run the selected configuration until the last epoch $T$.

2. **BOHB** [14] and **DEHB** [4]: We follow the most recent implementation of these algorithms in [3]. We slightly modify the official code (`https://github.com/releaunifreiburg/QuickTune`), which is based on SyneTune [46] package.

3. **DPL** [24]: We follow the official code (`https://github.com/releaunifreiburg/DPL`), and slightly modify the benchmark implementation to incorporate our experimental setups.

4. **ifBO** [44]: We use the official code (`https://github.com/automl/ifBO`) for the surrogate model (*i.e.*, the LC extrapolator), and incorporate it into our code base to be aligned with our experimental setups.

5. **DyHPO** [62] and **Quick-Tune**† [3]: We follow the official code (`https://github.com/releaunifreiburg/DyHPO`), and slightly modify the benchmark implementation to incorporate our experimental setups. For Quick-Tune†, we pretrain the deep kernel GP for 50,000 iterations with Adam optimizer with batch size of 512. The initial learning rate is set to $10^{-3}$ and decayed with cosine scheduling. To leverage the transfer learning scenario, we use the best configuration among the LC datasets which is used for training the GP as an initial guess of BO.

6. **FSBO** [60]: We use the official code (`https://github.com/releaunifreiburg/fsbo`), slightly modify it to incorporate our experimental setups, and use the best configuration among the LC datasets as an initial guess.

## G  Additional Experiments

Table 8: **Results on object detection datasets** ($c = 1$ and $\alpha \in \{0, 2^{-6}, 2^{-4}\}$). We multiply 100 to the normalized regret for better readability.

| Method | $\alpha = 0$ | | $\alpha = 2^{-6}$ | | $\alpha = 2^{-4}$ | |
|---|---|---|---|---|---|---|
| | Regret | Rank | Regret | Rank | Regret | Rank |
| Random | $5.0_{\pm 1.3}$ | 6.5 | $7.1_{\pm 2.6}$ | 6.4 | $13.1_{\pm 2.6}$ | 6.5 |
| BOHB | $3.2_{\pm 1.0}$ | 5.2 | $4.8_{\pm 1.0}$ | 5.3 | $10.7_{\pm 1.0}$ | 5.4 |
| DEHB | $5.0_{\pm 1.4}$ | 6.6 | $6.6_{\pm 1.4}$ | 6.5 | $12.4_{\pm 1.3}$ | 6.6 |
| DyHPO | $16.0_{\pm 2.5}$ | 5.9 | $17.5_{\pm 2.5}$ | 6.0 | $23.1_{\pm 2.7}$ | 6.2 |
| DPL | $3.9_{\pm 1.4}$ | 4.6 | $5.5_{\pm 1.4}$ | 4.8 | $11.4_{\pm 1.3}$ | 5.2 |
| ifBO | $2.3_{\pm 0.5}$ | 4.3 | $3.9_{\pm 0.5}$ | 4.3 | $9.8_{\pm 0.5}$ | 4.4 |
| Quick-Tune† | $5.3_{\pm 0.0}$ | 4.8 | $6.9_{\pm 0.0}$ | 4.9 | $12.6_{\pm 0.0}$ | 5.0 |
| FSBO | $2.1_{\pm 0.0}$ | 3.9 | $3.7_{\pm 0.0}$ | 3.9 | $9.6_{\pm 0.0}$ | 4.2 |
| **CFBO (ours)** | $\mathbf{1.3_{\pm 0.1}}$ | **3.3** | $\mathbf{3.6_{\pm 0.3}}$ | **2.9** | $\mathbf{5.7_{\pm 0.3}}$ | **1.4** |

**Effectiveness on the real-world HPO.** We investigate the effectiveness of CFBO on real-world object-detection dataset. From the 10 different datasets in RoboFlow100 [11], we collect 500 LCs of validation performances by training three different network architectures, such as ResNet-50 [20], HRNet [55], MobileNetv2 [47], with 4 different hyperparameters (`batch_size`, `learning_rate`, `momentum`, and `weight_decay_factor`). Based on this setting, we construct 30 tasks (=3 network architectures×10 datasets) and split them into 20/10 tasks for meta-training/meta-test, respectively. In Tab. 8, we observe that CFBO consistently and significantly outperforms all baselines on this real-world dataset.

**Visualization of normalized regret.** We provide visualization for the normalized regret of each method on LCBench, TaskSet, and PD1 throughout Figs. 11 to 16.

**Visualization of the LC extrapolation over BO steps.** We provide visualization for the LC extrapolation over BO steps of CFBO throughout Figs. 17 to 22. Specifically, we plot the LC extrapolation results of *unseen* hyperparameter configurations through BO process. Each row shows the results for a different size of the observation set ($|\mathcal{C}| \in \{0, 10, 50, 300\}$), and each column shows a different size of context points ($y_{n, \leq t}$, where $t \in \{0, 2, 5, 10, 20, 30\}$) in each LC.

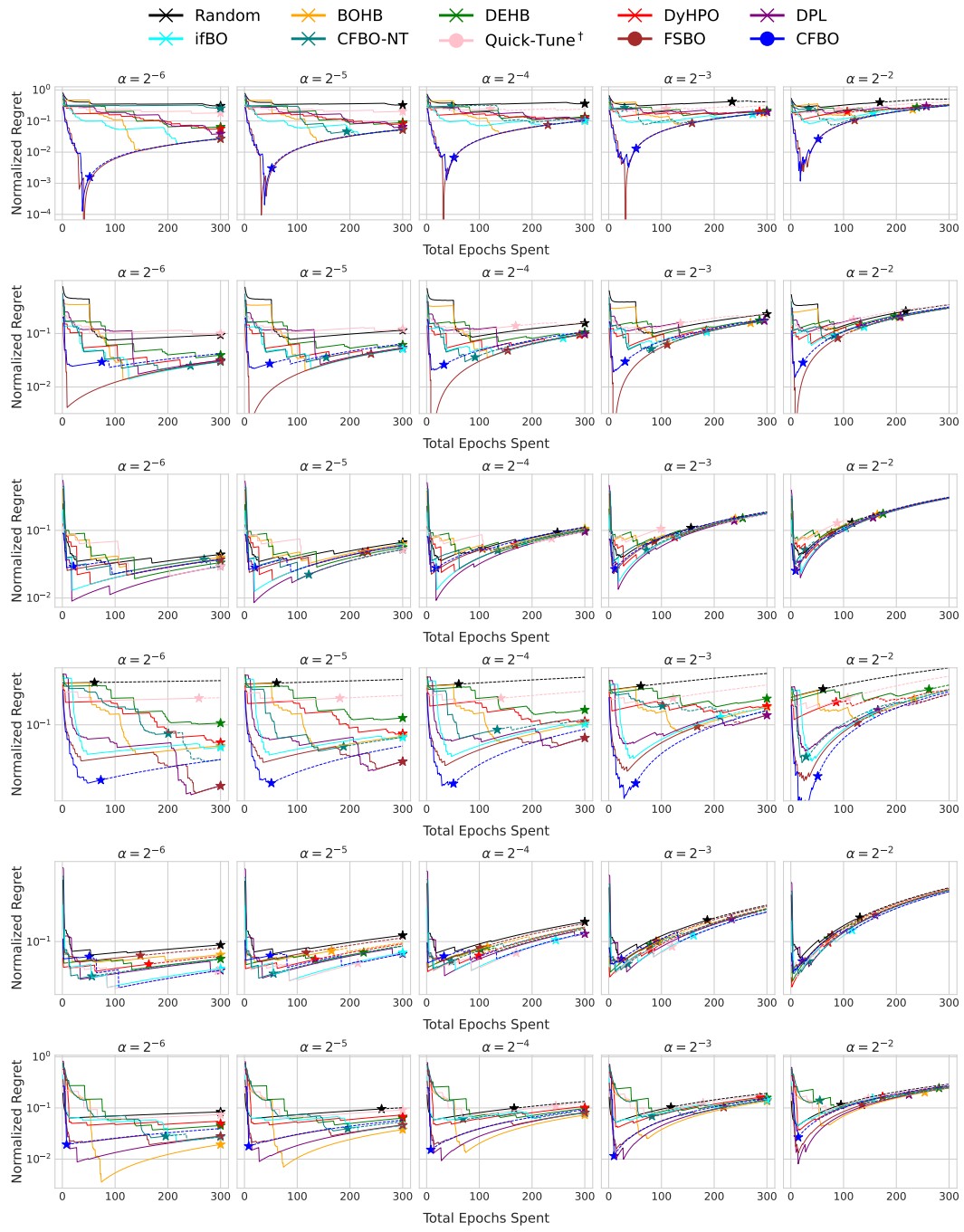

Figure 11: Visualization of the **normalized regret on LCBench**.

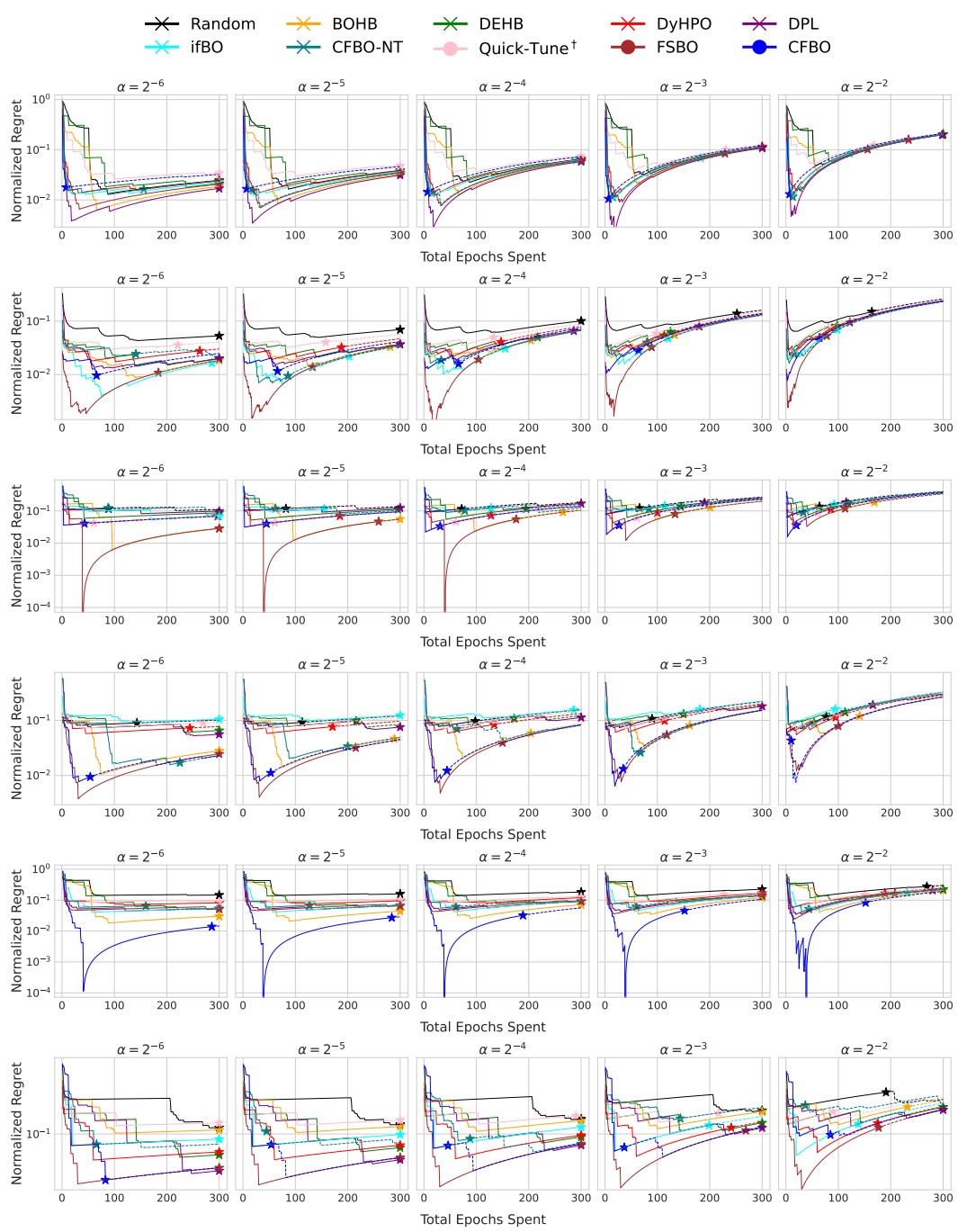

Figure 12: **Visualization of the normalized regret on LCBench**.

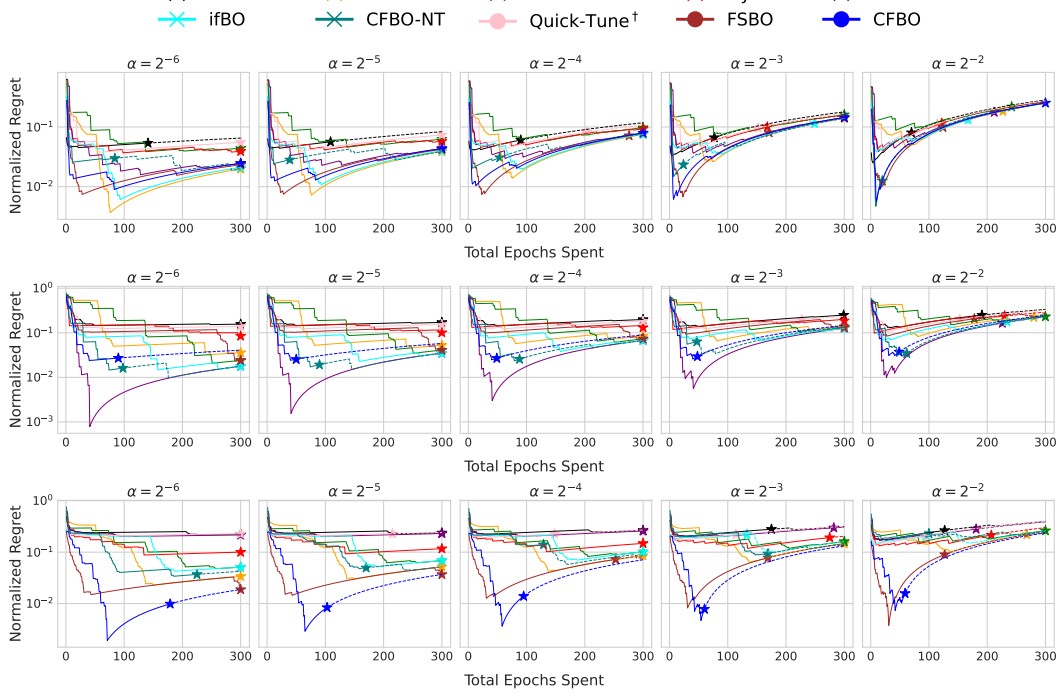

Figure 13: Visualization of the **normalized regret on LCBench**.

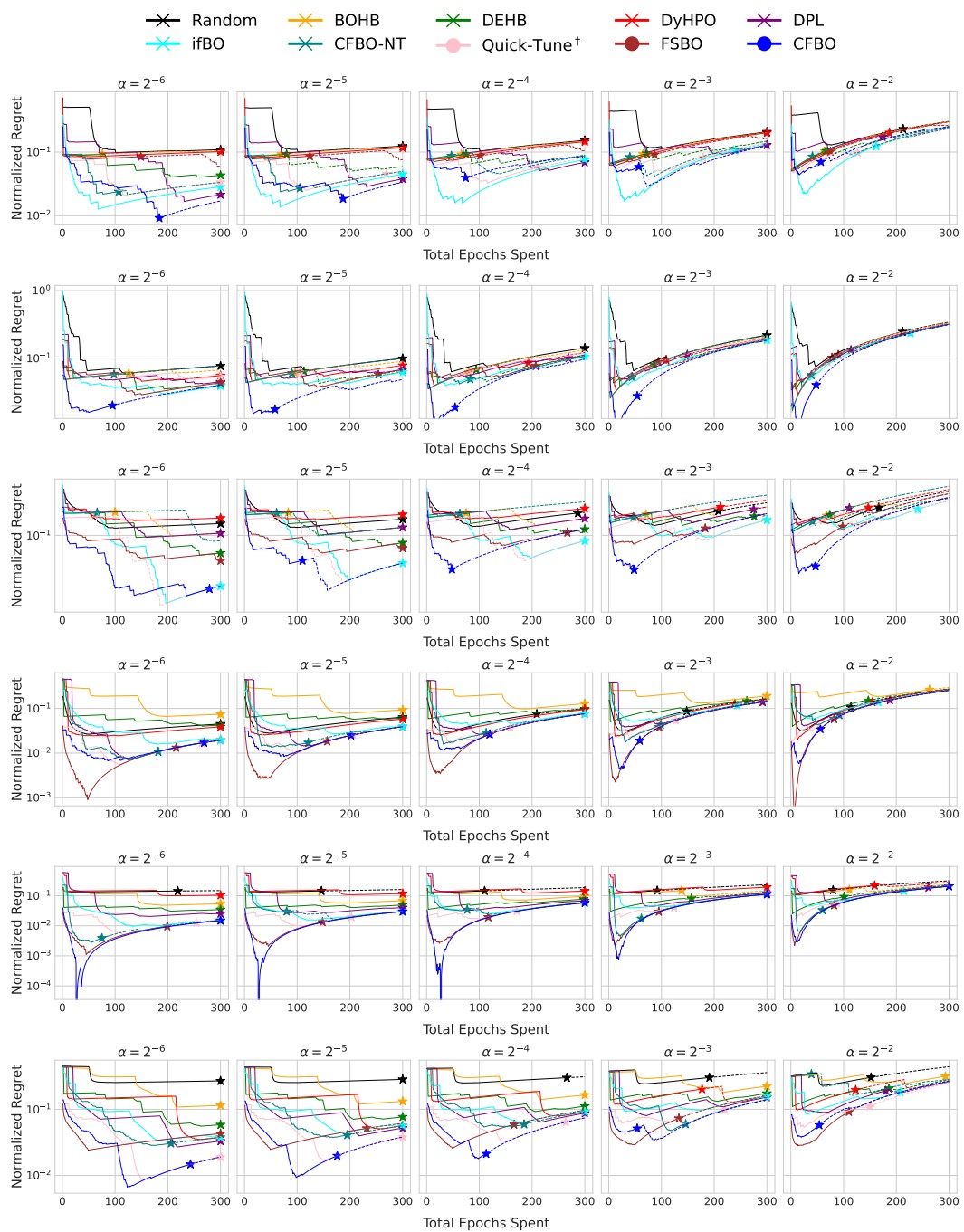

Figure 14: Visualization of the **normalized regret on TaskSet**.

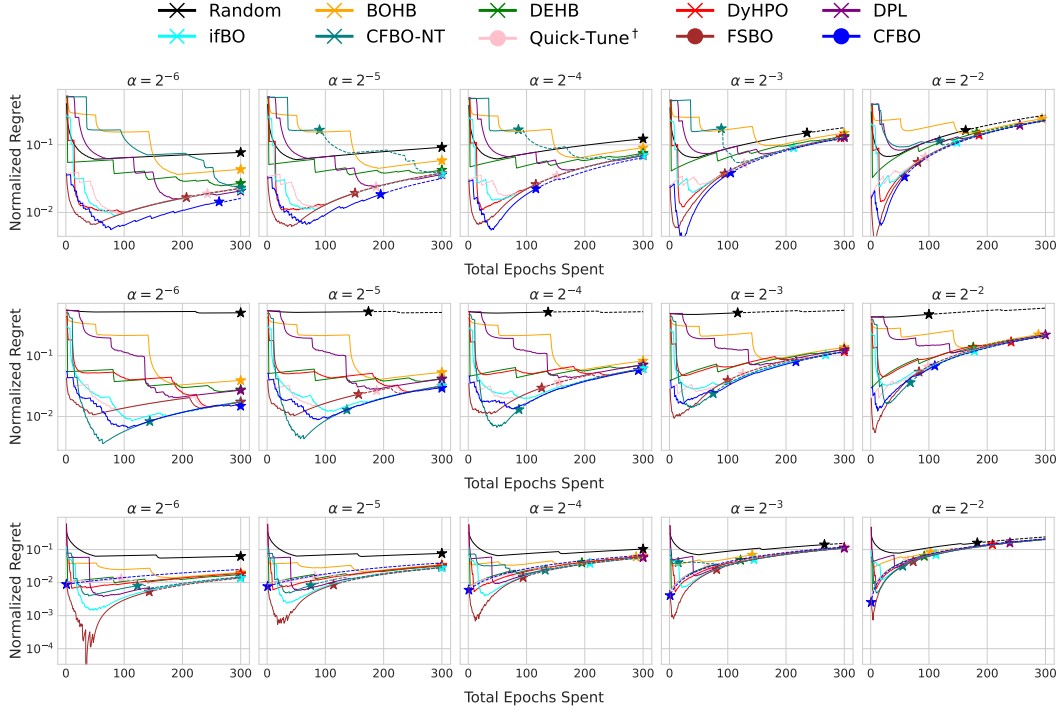

Figure 15: Visualization of the **normalized regret on TaskSet**.

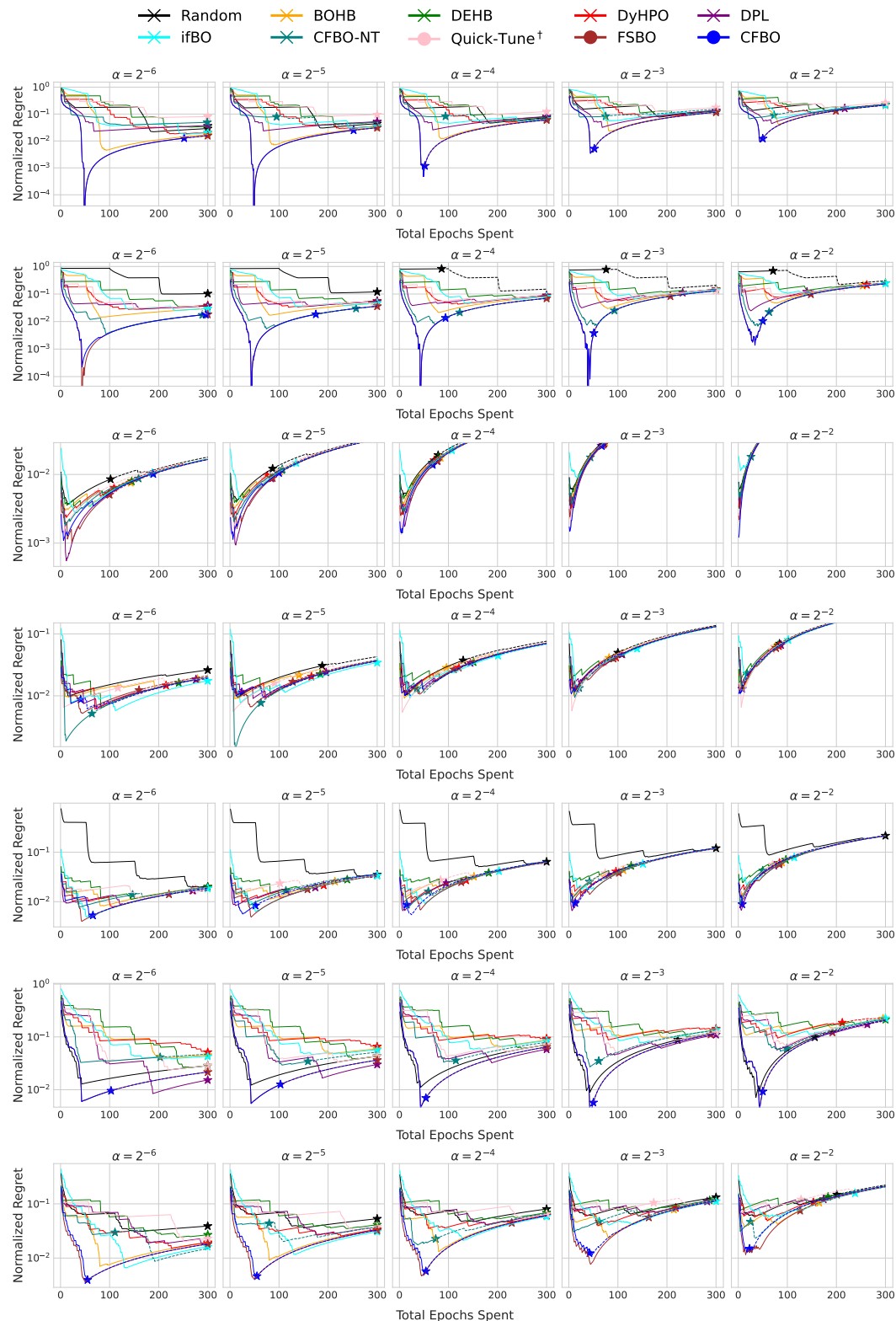

Figure 16: Visualization of the **normalized regret on PD1**.

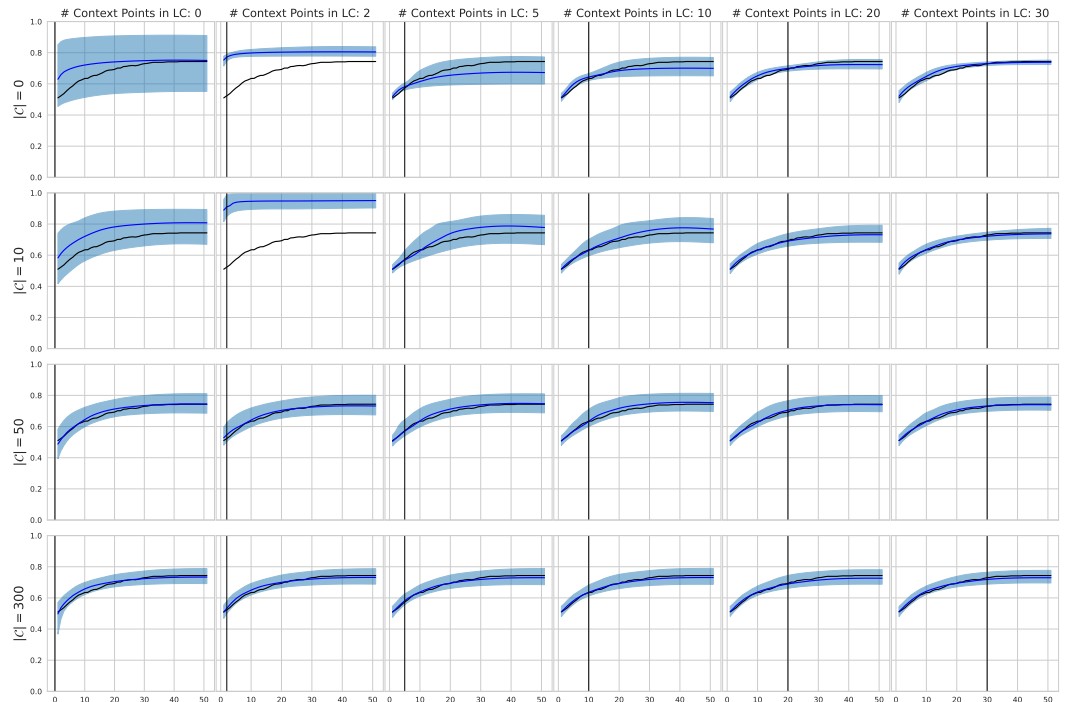

Figure 17: Visualization of **LC extrapolation on LCBench**.

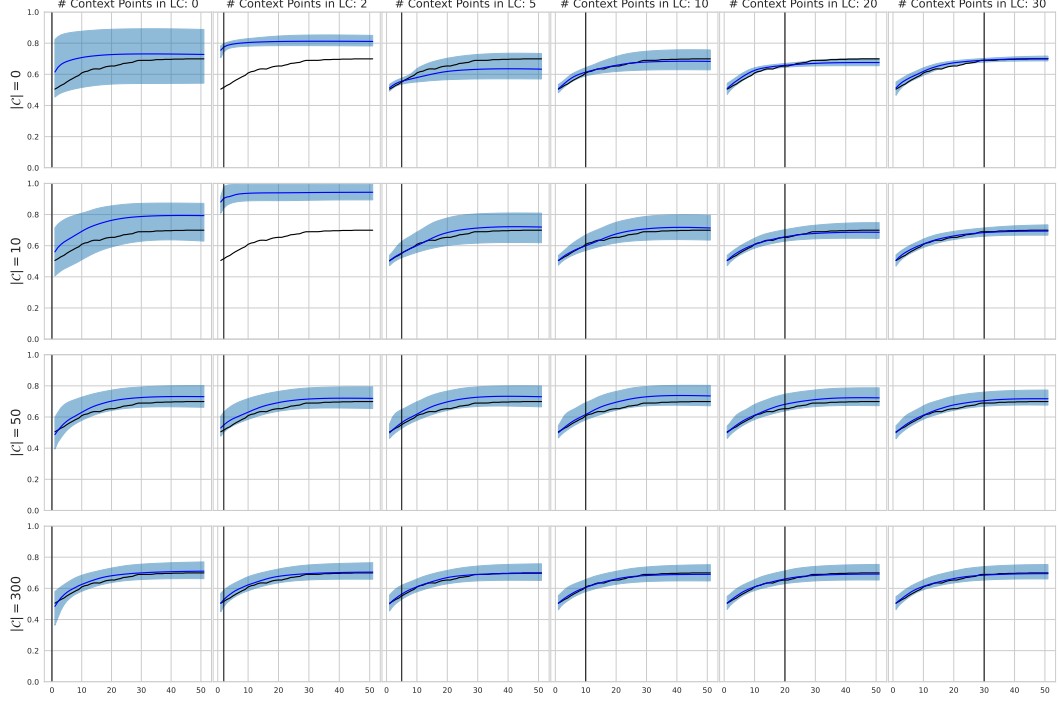

Figure 18: Visualization of **LC extrapolation on LCBench**.

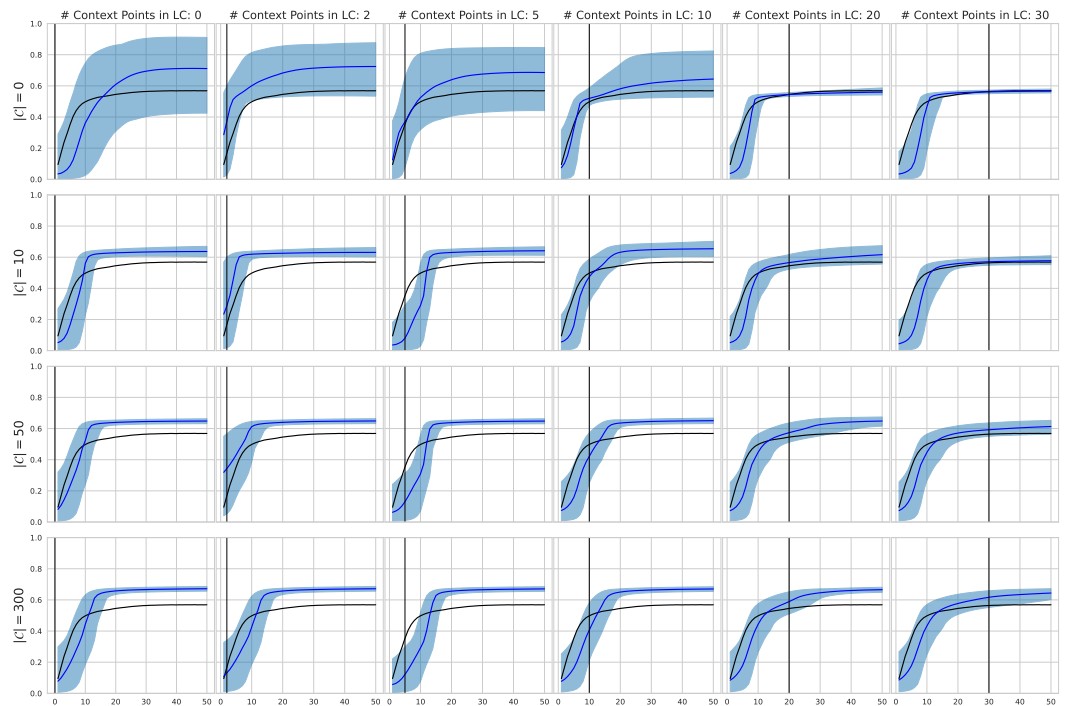

Figure 19: Visualization of **LC extrapolation on TaskSet**.

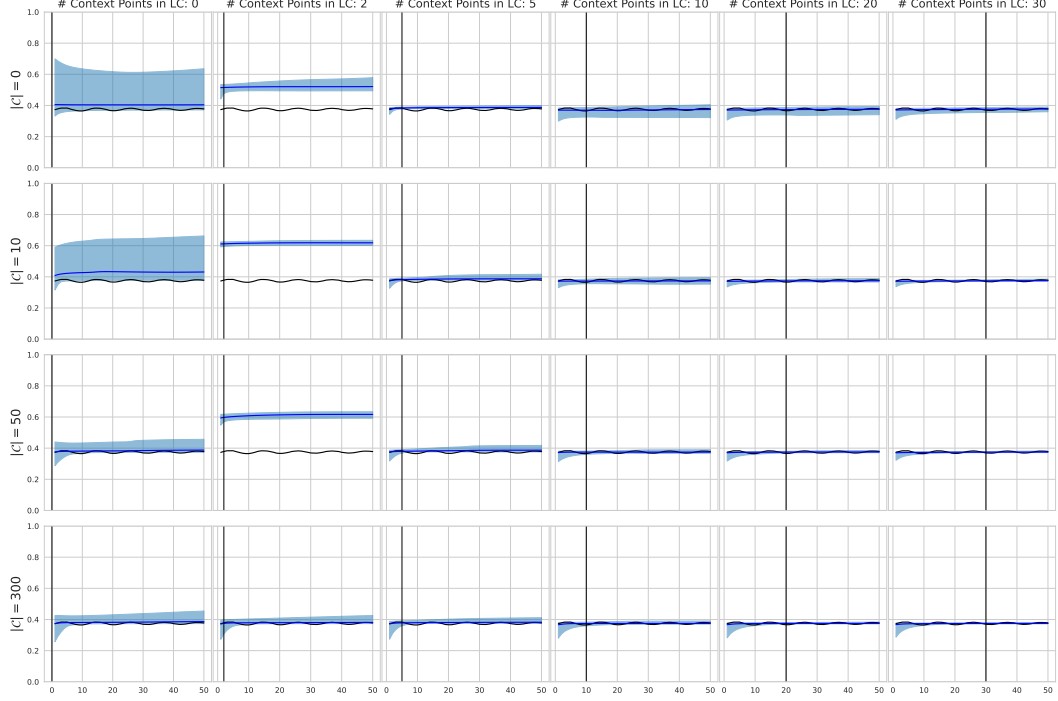

Figure 20: Visualization of **LC extrapolation on TaskSet**.

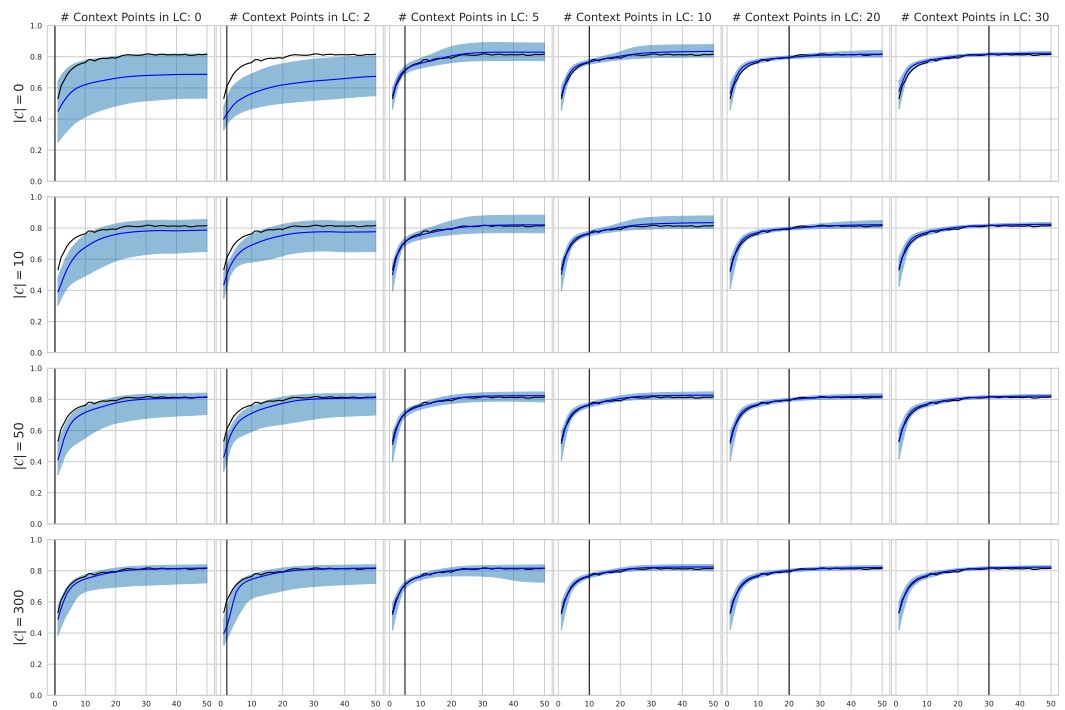

Figure 21: Visualization of **LC extrapolation on PD1**.

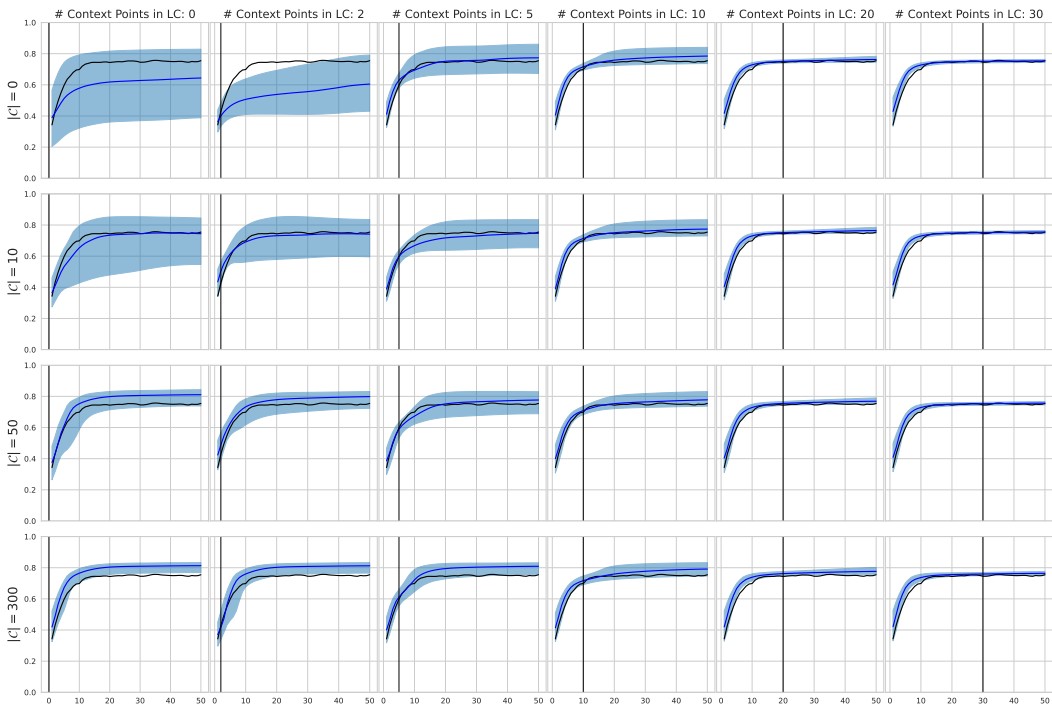

Figure 22: Visualization of **LC extrapolation on PD1**.

