# OpenReview forum: "Cost-Sensitive Freeze-thaw Bayesian Optimization for Efficient Hyperparameter Tuning"
_NeurIPS.cc/2025/Conference — NeurIPS 2025 poster_

### Official Review · Reviewer_Vxb6 · 2025-06-24

**Clarity:** 2
**Significance:** 3
**Originality:** 3
**Rating:** 5
**Confidence:** 3

**Summary:**

The paper describes a freeze-thaw BO algorithm for hyper-parameter optimization that optimizes the utility, which is a measure of the user's internal tradeoff between cost and performance.

**Questions:**

See above re my (mis)understanding of equation (2).

**Ethical Concerns:**

["NO or VERY MINOR ethics concerns only"]

**Final Justification:**

My main point of uncertainty in the initial reviews was Q2. The authors response covers this, so I am happy to increase my recommendation to accept.

**Limitations:**

Yes.

**Quality:**

3

**Strengths And Weaknesses:**

The central idea behind this paper is solid and the paper is well-written for the most part. The paper is well structured - relevant background material is introduced systematically, the algorithm described clearly and the experimental verification appears reasonable.

It strikes me that clarity has been somewhat compromised to fit within the page limit in some places. For example I am not at all familiar with PFNs, so ideally I would like a more detailed exposition to be presented: however I understand that this may be incompatible with page limits. Also the figures in the results section are basically unreadable, but again I understand that this is almost certainly due to page limits, so I will overlook this.

More seriously equation (2) is not clear to me. If I understand correctly, $u_1$ and $u_2$ are pairs $(b,\tilde{y}_n)$, which appears incompatible with exponentials and inversions as per the equation. This seems central to the paper - could you please clarify?

---

> ### Author Rebuttal · Authors · 2025-07-30
>
> We sincerly appreciate your constructive comments to improve our paper, and the positive review. We address your concerns in the following:
>
> ---
>
> >**[Q1]** It strikes me that clarity has been somewhat compromised to fit within the page limit in some places. For example I am not at all familiar with PFNs, so ideally I would like a more detailed exposition to be presented: however I understand that this may be incompatible with page limits.
> - We sincerely apologize for the insufficient explanation of essential preliminaries.
> - To improve clarity and ensure self-containedness, we include a more detailed exposition of the PFN-based learning curve extrapolation mechanism in `Appendix D`.
> - We will revise the main text to include a pointer:
>   "Please refer to Appendix D for the detailed mechanism of learning curve extrapolation."
>
>
> >**[Q2]** Also the figures in the results section are basically unreadable, but again I understand that this is almost certainly due to page limits, so I will overlook this.
> - We sincerly apologize you about the unreadbility of experimental results, espeically `Figure 6`.
> - We ensure you that we will try our best to improve the clarity of experimental results in the revision by making room for the figures.
>
> ---
>
> >**[Q2]** More seriously equation (2) is not clear to me. If I understand correctly, $u_1$ and $u_2$ are pairs $(b, \tilde{y}_n)$, which appears incompatible with exponentials and inversions as per the equation. This seems central to the paper - could you please clarify?
>
> - We sincerely apologize for the confusion. `Eq. 2` defines the objective for fitting the utility function using **preference learning** via the Bradley–Terry model [1].
> - In this setup, we are given a pair of outcomes—e.g., $(b_1, y_1)$ and $(b_2, y_2)$, where $b_1, b_2 \in [B]$ and $y_1, y_2\in[0,1]$—and a corresponding preference label indicating which is preferred.
> - Let $u_1 := U(b_1, y_1) \in [0,1]$ and $u_2 := U(b_2, y_2) \in [0,1]$ denote the utility scores of these two outcomes. These are **scalar** values derived from a parametric utility function.
> - The Bradley–Terry model is then trained, i.e., minimize `Eq. 2` w.r.t the parameters of  $U$ (e.g., $\alpha$), on such pairs $(u_1, u_2)$ with preference labels ($u_1>u_2$), and the exponential/logistic form in `Eq. 2` applies directly to these scalar utilities—not the raw pairs.
> - We will clarify this in the revision.
>
> ---
>
> ### References
>
> [1] Bradley, Ralph Allan, and Milton E. Terry. "Rank analysis of incomplete block designs: I. the method of paired comparisons." Biometrika. 1952.

---

> > ### Comment · Reviewer_Vxb6 · 2025-08-05
> >
> > Thank you for your response, in particular to Q2. After reviewing the paper again it seems to me that there is a sufficient contribution here to raise my recommendation slightly.

---

> > > ### Author Response · Authors · 2025-08-06
> > >
> > > We sincerely appreciate your decision and assure you that we will clarify the above discussion in the revision, particularly our explanation of **[Q2]**.

---

### Official Review · Reviewer_Ly5J · 2025-06-24

**Clarity:** 3
**Significance:** 2
**Originality:** 2
**Rating:** 3
**Confidence:** 4

**Summary:**

The paper introduces a cost-sensitive freeze-thaw BO method for HPO (CFBO) that focuses on performance vs. computational cost. The core idea is optimizing a utility function defined on performance and cost.
This utility function in general can be user-defined.
CFBO is built upon ideas from freeze-thaw BO, however, the acquisition function is based on expected utility improvement instead of performance to (dynamically) determine the training length of configurations with high expected utility (performance vs. computational cost trade-off).
The authors further propose a regret-based stopping rule defined on the utility to terminate optimization when further improvements become inefficient.
The learning curve extrapolation within the freeze-thaw part makes use of PFNs and builds on existing works but implements a mixup-based transfer learning strategy.
In their experiments, the authors compare CFBO against baselines and competitors on three multi-fidelity HPO benchmarks and show that CFBO outperforms all baselines in cost-sensitive settings when looking at the regret in utility.
In conventional HPO with no cost penalty, CFBO performs competitively.

**Questions:**

I already reviewed the paper this year when it was submitted to ICML 2025.
As far as I can tell, only minor changes were made prior to re-submitting this version.
I therefore follow up with similar questions I already had when first reviewing:

How did you select the maximum budget for each benchmark task?

The expected improvement of utility in 2.2 has the implicit assumption that past performance (and utility) observations are noise-free, correct?
I.e., in (3), we subtract from the utility function at a future time $b + \Delta t$ with cumulative best performance $\tilde{y}_{b + \Delta t}$ the most recent utility value $U_p$.
However, the expectation only applies to the utility part of the future time and is derived from the predicted learning curve and $U_p$ is treated as deterministic.

In the checklist regarding limitations you state:
`Justification: We provide the Limitations paragraph in §4.`
However, this simply references the Conclusion section were limitations are not discussed.
Can you discuss this in the context of HPO where the objective function is per se not deterministic (i.e., in contrast to the tabular benchmarks with pre-collected data used in the experiments, real-world HPO is stochastic and noisy as we are optimizing an estimate of the generalization performance obtained via model fitting and evaluation based on a resampling)?

I believe it could be beneficial to add a notation table in the appendix.

**Ethical Concerns:**

["NO or VERY MINOR ethics concerns only"]

**Final Justification:**

I do maintain a somewhat reserved stance but can acknowledge several improvements in my understanding of the paper's contributions.
The authors have successfully clarified the practical motivation for cost-sensitive HPO (e.g. in cloud computing settings) and addressed runtime concerns by pointing to negligible overhead compared to model training time.

However, for me some concerns remain.
While the problem formulation is interesting, the technical approach still remains largely incremental, primarily combining existing methods in a building block fashion, although with some implementation challenges or minor technical challenges that needed to be solved.
Moreover, the majority of evaluation setting in the paper give CFBO an inherent advantage as the only method naturally operating in the utility-aware setting.

For me, of concern is also that previously raised issues about connections to early stopping literature were not addressed in this submission, despite the paper being previously reviewed (ICML) with similar feedback.
While the authors have now promised to emphasize this related work more strongly in the appendix, I believe this literature is important enough to be included in the main paper.
The authors showed some progress during the rebuttal by implementing FSBO+[8], but the results were (somewhat unexpectedly) poor.
This might be due to structural mismatches (the method not operating in the multi-fidelity setting like CFBO, or because the benchmarks only provide simple validation holdout splits, making it challenging to apply early stopping techniques that rely on variance estimates of generalization error without further adjustments).

Therefore, while I have slightly increased my score, I cannot actively push for acceptance.
Overall, after reading the other reviews and the author responses, I do not have a strong opinion about rejection either because the paper does have its pros and therefore I maintain this borderline rating.

**Limitations:**

Limitations and potential negative societal impact are addressed adequately.

**Paper Formatting Concerns:**

There are no obvious formatting issues.

**Quality:**

3

**Strengths And Weaknesses:**

### Strengths

CFBO makes use of solid building blocks to tackle the problem of cost-sensitive multi-fidelity HPO (e.g., freeze-thaw BO, transfer learning for learning curve extrapolation via PFNs).
Benchmarks are reasonably selected and the set of competitors and baselines is sufficiently large.
The authors investigate normalized regret (on utility) and rank for each benchmark.
The paper is written well and clarity is high.
Originality is given to some extent and the problem CFBO tackles is a relevant one.
CFBO ties in with the related literature of multi-fidelity HPO, focusing on freeze-thaw BO.
The authors correctly identify that existing methods generally do not focus on performance vs. computational cost trade-offs but instead assume a fixed budget.



### Weaknesses

Novelty is limited given the reuse of existing building blocks (EI, freeze-thaw BO, learning curve extrapolation via PFNs).
One notable caveat is that all other baselines and competitors are not naturally designed to consider the trade-off between performance and computational cost so the metric of normalized regret on utility can be considered slightly biased w.r.t CFBO.
Computational efficiency is not analyzed in detail and detailed wall-clock time comparisons are missing.
Overall, the method is strongly heuristically motivated.

The authors do not address recent literature concerned with terminating BO ([1], [2]).
Especially [1] is directly tailored to the problem of HPO and stopping is performed if an upper bound on the simple regret (difference between current best validation score and estimated global minimum based on the surrogate model) is less or equal to the estimated variance of the validation error.
This stopping criterion is less heuristically motivated than the one proposed by the authors and further does take uncertainty in the objective function in HPO into account.
In l738 in the appendix, the authors briefly address [1] but I do not think that this is sufficient.
Moreover, the paper does not make a strong connection to the general literature of cost-aware BO [3].

[1] Makarova et al. (2021) https://arxiv.org/abs/2104.08166

[2] Wilson (2024) https://arxiv.org/abs/2402.16811

[3] Xie et al. (2024) https://arxiv.org/abs/2406.20062

---

> ### Author Rebuttal · Authors · 2025-07-30
>
> We sincerly appreciate you for your time and constructive comments which improve our paper. We respectfully address your concerns as following:
>
> ---
> >**[Q1-1]** Limited novelty.
> - We respectfully disagree with this assessment. While freeze–thaw BO, transfer learning, and preference learning have appeared in prior work, CFBO is novel in **how it combines these elements** to address a **new problem**: hyperparameter optimization when users **explicitly trade off model performance against computational cost**.
> - As discussed in `L32–36`, cloud‑service users often work under credit budgets—they care about performance but also penalize HPO cost to conserve resources. Existing methods **either ignore cost or treat it as a hard constraint**, rather than optimizing expected utility. We show that traditional methods perform poorly under such trade-offs.
> - CFBO is designed for this realistic setting. We (1) formulate a **different notion of cost‑aware HPO problem**, (2) define a **utility‑based objective**, (3) explain why **prior methods fall short**, and (4) integrate known tools into **an effective solution**.
> - The contribution lies not just in technical components but in defining and solving this practically motivated problem. We will clarify this in the revision.
> ---
> >**[Q1-2]** Baselines not designed for trade-off.
> - Thank you for pointing this out. We already provide a fair comparison under the conventional multi‑fidelity HPO setting with $\alpha=0$ (`Figure 6`), where utility reduces to pure performance. CFBO still outperforms all baselines, showing its advantage is **not due to privileged knowledge**.
> - Incorporating our utility into existing baselines is **fundamentally difficult**. $U$ depends on future performance–cost trade‑offs and requires full learning curve inference, which DyHPO [1] and FSBO [2] cannot support as they predict only the next or final step.
> - For fairness, we tuned a fixed stopping threshold ($\delta=0.2$) for all baselines. They cannot use our adaptive threshold $\delta_b$ since it relies on computing future utility (`Eq. 6`).
> - We also implemente **CFBO‑NT** by augmenting ifBO [3] with our stopping and acquisition rules. It outperforms all baselines without transfer learning, confirming that gains stem from **our methodological contributions**.
> - We will include this in the revision.
> ---
> >**[Q1‑3]** Missing runtime analysis.
> - We kindly note that in `L897‑907` of `Appendix E`, we report wall‑clock times for running a full BO process (300 epochs) with DPL, IFBO, and CFBO on LCBench, TaskSet, and PD1.
> - While IFBO is the most efficient, the additional cost of CFBO is **negligible** because model training dominates total runtime.
> - In real‑world HPO experiments (`Appendix E`), we train ResNet‑50 [4], HRNet [5], and MobileNetV2 [6] on 10 RoboFlow100 [7] datasets to collect 500 curves.
> - Across these runs, CFBO adds only 44.4 s to 281.6 s relative to IFBO—trivial compared to training modern networks for 300 epochs.
> ---
> >**[Q1-4]** Heuristic motivation.
> - Thank you for pointing this out. While some components (e.g., $\text{BetaCDF}$ interpolation and adaptive stopping) are heuristic in form, they are derived from an **explicit utility‑maximization objective** and guided by uncertainty estimates over learning curves. `Table 1` provides ablations that isolate each component and confirm its quantitative impact.
> - We also evaluate CFBO under diverse settings, from conventional HPO ($\alpha=0$) to three utility functions with different penalties ($c\in\\{1,2,0.5\\}$ and $\alpha\in\\{2^{-6},\ldots,2^{-2}\\}$), validating the heuristics at well‑understood extremes. Across four benchmarks (including the experiment in `Appendix F`), CFBO consistently outperforms baselines, demonstrating that these heuristics translate into reliable real‑world gains.
> ---
> > **[Q2‑1]** Missing discussion of BO stopping works [8,9].
> - Thank you for this concern. **We respectfully explain below how the stopping rule of Makarova et al. [8] roughly fits within our utility‑based framework**.
> - Proposition 1 in [8] states that, for BO step $b$,$$f^* - f_b \le 2\epsilon + \tilde{y}^* - \tilde{y}_b,$$
> - where $f$ denotes tbe true population performance, $\tilde{y}_b$ the validation performance, $*$ a maximizer, and $\epsilon$ statistical error. The above leads to the **stopping condition** $\tilde{y}^\* -\tilde{y}_b\le\epsilon$.
> - This can be expressed through a utility:$$U(b,\tilde{y}_b)=\begin{cases}\tilde{y}_b&\text{if }\tilde{y}_b \ge\tilde{y}^* -\epsilon,\\\ -\infty &\text{otherwise}.\end{cases}$$
> - Since $\tilde{y}^\*$ and $\epsilon$ are unknown, [8] estimates them via LCB/UCB and CV variance. This estimator is a function of $b$, the above can be rewritten as:$$U(b,\tilde{y}_b)=\begin{cases}\tilde{y}_b&\text{if }\tilde{y}_b \ge c(b),\\\ -\infty &\text{otherwise}.\end{cases}$$
> - Interpreted this way, the preference in [8] is **"stop once performance exceeds $c(b)$; additional budget has no value thereafter."** This represents a special case of our utility formulation.
> - Wilson [9] similarly fits into a utility view:$$U(b,\tilde{y}_b)=\begin{cases}\tilde{y}_b &\text{if } Pr[r_b\le \epsilon|\mathcal{C}]<1-\delta,\\\ - c_b &\text{otherwise},\end{cases}$$where $r_b=f^*-f_b$ and $c_b$ is cumulative cost.
> - We will revise `Appendix A` to incorporate these links.
> ---
> >**[Q2-2]** Link to cost‑aware BO [10].
> - Xie et al. [10] also pursue a cost‑sensitive objective, but their Gittins‑index policy is designed for single‑fidelity black‑box BO, whereas our method predicts full learning curves and uses an adaptive utility‑based stopping rule in the multi‑fidelity HPO setting. We will add this clarification to `Appendix A`.
> ---
> >**[Q3]** Choice of maximum budget.
> - In the multi‑fidelity BO literature there is no fixed rule for the maximum budget, though many studies adopt 1000 epochs/evaluations.
> - We set the cap to 300 to reflect cloud‑credit limits and highlight early stopping. CFBO‑NT still outperforms baselines, showing gains are not solely from transfer learning.
> ---
> >**[Q4]** EI assumes noise‑free past utility $U_p$.
> - Yes, we follow the classical Expected Improvement assumption: the best utility observed so far, $U_p$, is treated as a fixed constant, and uncertainty is modeled only for the future term $U(b+\Delta t,\tilde{y}_{b+\Delta t})$.
> - If $U_p$ were noisy, one could adopt noisy‑EI variants that integrate over its posterior [11,12]; we keep the deterministic form for simplicity.
> ---
> >**[Q5]** Limitations paragraph missing.
> - We will add a **Limitations** subsection in `§4`. Our PFN‑based extrapolator works on discrete steps and may falter with continuous budgets and noise; observation noise can destabilise utility and stopping. We will explicitly acknowledge these points.
> ---
> >**[Q6]** Need notation table.
> - We sincerely agree with this suggestion and will include the provided notation table in `Appendix`.
>
> |Notation|Description|
> |-|-|
> |$x_n\in\mathbb{R}^{d_x}$|$n$‑th hyperparameter configuration|
> |$\mathcal{X}=\\{x_1,\ldots,x_N\\}$|Set of hyperparameter configurations|
> |$T\in\mathbb{N}$|Maximum training epochs for each configuration|
> |$t\in[T]:=\\{1,\ldots,T\\}$|Training‑epoch index|
> |$y_{n,t}\in[0,1]$|Performance of $x_n$ at epoch $t$|
> |$B\in\mathbb{N}$|Maximum total training epochs (overall BO budget)|
> |$b\in[B]$|Budget spent so far|
> |$\tilde{y}_b$|Best performance observed up to budget $b$|
> |$\mathcal{C}=\{(x,t,y)\}$|Collected partial or full learning curves|
> |$q_\theta:\mathbb{R}^{d_x}\times[T]\times\mathcal{C}\to[0,1]$|Learning‑curve extrapolator|
> |$U:[B]\times[0,1]\to\mathbb{R}_{\ge0}$|Utility function|
> |$U_p\in\mathbb{R}_{\ge0}$|Utility immediately before current BO step|
> |$A(\cdot;U):[N]\to\mathbb{R}$|Acquisition function (`Eq. 3`)|
> |$\hat{U}_{\text{max}}$|Maximum utility observed so far|
> |$\hat{U}_{\text{min}}:=U(B,\tilde{y}_1)$|Approximated minimum utility|
> |$\hat{R}\_b:=\frac{\hat{U}_{\text{max}} - U_p}{\hat{U}\_\text{max} - \hat{U}\_\text{min}}\in [0,1]$|Normalized regret|
> |$\delta\in[0,1]$|Fixed stopping threshold for baselines|
> |$\delta_b:=\text{BetaCDF}(p_b,\beta,\beta)^\gamma\in[0,1]$|Adaptive stopping threshold|
> |$p_b\in[0,1]$|Probability that current configuration will improve $U_p$ (Eq. 6)|
> |$\text{BetaCDF}(\cdot;\beta,\beta):[0,1]\to[0,1]$|Beta CDF with shape $\beta$|
> |$\beta\in\mathbb{R}_{>0}$|Hyperparameter controlling interpolation|
> |$\gamma\in\mathbb{R}_{>0}$|Hyperparameter setting $\delta_b$ at $p_b$|
> ---
> ### References
> [1] Wistuba, Martin, Arlind Kadra, and Josif Grabocka. "Supervising the multi-fidelity race of hyperparameter configurations." NeurIPS. 2022.
>
> [2] Wistuba, Martin, and Josif Grabocka. "Few-shot Bayesian optimization with deep kernel surrogates." arXiv. 2021.
>
> [3] Rakotoarison, Herilalaina, et al. "In-context freeze-thaw bayesian optimization for hyperparameter optimization." ICML. 2024.
>
> [4] He, Kaiming, et al. "Deep residual learning for image recognition." CVPR. 2016.
>
> [5] Wang, Jingdong, et al. "Deep high-resolution representation learning for visual recognition." IEEE transactions on pattern analysis and machine intelligence. 2020.
>
> [6] Sandler, Mark, et al. "Mobilenetv2: Inverted residuals and linear bottlenecks." CVPR. 2018.
>
> [7] Ciaglia, Floriana, et al. "Roboflow 100: A rich, multi-domain object detection benchmark." arXiv. 2022.
>
> [8] Makarova, Anastasia, et al. "Automatic termination for hyperparameter optimization." International Conference on Automated Machine Learning. PMLR, 2022.
>
> [9] Wilson, James. "Stopping Bayesian optimization with probabilistic regret bounds." NeurIPS. 2024.
>
> [10] Xie, Qian, et al. "Cost-aware Bayesian optimization via the Pandora's Box Gittins index." NeurIPS. 2024.
>
> [11] Picheny, Victor, Tobias Wagner, and David Ginsbourger. "A benchmark of kriging-based infill criteria for noisy optimization." Structural and multidisciplinary optimization. 2023.
>
> [12] Letham, Benjamin, et al. "Constrained Bayesian optimization with noisy experiments." 2019.

---

> > ### Comment · Reviewer_Ly5J · 2025-08-04
> > **Response to Authors**
> >
> > Thank you for the detailed rebuttal which addresses many of my questions.
> > I do have some follow-up points:
> >
> > [Q1-1] Regarding novelty, I acknowledge your argument that novelty can show itself in different ways, including problem framing and potentially also combinations of existing methods to solve a problem.
> > While I can see merit in your contribution from this perspective, I do maintain that the technical novelty is somewhat limited due to the heavy reliance on existing building blocks for the core components and this is why I stated limited novelty in the review.
> >
> > [Q1-3] Regarding runtime analysis, while I appreciate the pointer to Table 5 in Appendix E showing wallclock times, I have two questions: The current analysis provides only a limited view of computational overhead. Given the multi-fidelity nature of the problem, it would be great to see anytime performance curves with wallclock time (including overhead) on the x-axis. Regarding the real-world experiments: Could you provide runtime measurements for these experiments? Table 6 only shows regret and rank, and it would be helpful to understand the total wallclock time (and BO overhead) for these more realistic scenarios.
> >
> > [Q2-1] Regarding stopping criteria, I find your attempt to frame existing stopping mechanisms within your utility framework interesting. However, I do have two comments: I am not fully convinced that the mapping between the stopping mechanisms is as straightforward as suggested (and 1 to 1), particularly for [8] which operates under different assumptions about optimization and budget trade-offs. Moreover, given that the stopping criterion of CFBO is presented as one of your contributions, the paper would benefit from an explicit empirical comparison between CFBO's stopping mechanism and existing approaches (e.g., [8]). I do believe this would be helpful to show the advantages of your proposed method in practice while also making the connection to this related literature of early stopping BO.

---

> ### Author Response · Authors · 2025-08-06
>
> We sincerely appreciate you for **your efforts and participation on this discussion period**. We address your additional concerns below:
>
> ---
>
> > [Q1-1] Regarding novelty, I acknowledge your argument that novelty can show itself in different ways, including problem framing and potentially also combinations of existing methods to solve a problem. While I can see merit in your contribution from this perspective, I do maintain that the technical novelty is somewhat limited due to the heavy reliance on existing building blocks for the core components and this is why I stated limited novelty in the review.
>
> - We fully understand your concern and appreciate your acknowledgment. We **respectfully argue that our approach does more than simply combine existing blocks**, as detailed below.
>
> - Classic Expected Improvement (EI) optimizes expected performance only. We embed a **user-defined utility** that balances performance and cost directly into the EI objective, using Monte-Carlo estimation over learning-curve (LC) extrapolations. The resulting score (Eq. 6) doubles as a principled stopping rule. This unified, cost-sensitive EI is new for multi-fidelity BO and underpins the observed gains—for example, CFBO-NT markedly outperforms IfBO on cost-sensitive MF-HPO while using the same LC extrapolator.
>
> - We train the LC extrapolator (PFN) on **real LC data** instead of synthetic priors for stronger transfer (`§2.3`). The probelm is that real data are limited, therefore, Transformer-based PFNs can overfit quickly. To mitigate this, we introduce **LC-Mixup**, which blends two curves to create additional realistic samples. We observe LC-Mixup is crucial for the performance: on the PD1 dataset, the test negative log-likelihood rises sharply after 10k training iterations without it, and conventional MF-HPO performance ($\alpha=0$) degrades substantially. These results will be added in the revision.
>
> - Beyond our problem framing and combination of technical components, we strongly believe the above are newly introduced in the literature.
>
> ---
>
> > [Q1-3] Regarding runtime analysis, while I appreciate the pointer to Table 5 in Appendix E showing wallclock times, I have two questions: The current analysis provides only a limited view of computational overhead. Given the multi-fidelity nature of the problem, it would be great to see anytime performance curves with wallclock time (including overhead) on the x-axis. Regarding the real-world experiments: Could you provide runtime measurements for these experiments? Table 6 only shows regret and rank, and it would be helpful to understand the total wallclock time (and BO overhead) for these more realistic scenarios.
>
> - Thank you for pointing this out. We would first like to kindly note that our direct baseline, ifBO [1], also reports only the anytime performance curves with respect to the total number of training epochs (i.e., the number of BO observations), as shown in `Figure 3` and `Figure 4` of the paper. This is because a fundamental assumption in BO is that the number of observations is the primary factor determining performance.
>
> - Moreover, algorithm runtime or overhead is highly dependent on hardware. In particular, methods based on neural network-based LC extrapolators—such as DPL [2], ifBO, and CFBO—leverage GPU devices, and their overhead can be significantly reduced (to near-negligible levels) with better hardware (e.g., NVIDIA H200) or highly parallelized multi-GPU setups.
>
> - As a result, even without such runtime optimization, the relative computational overhead of different algorithms is negligible compared to the cost of training modern neural networks. For example, in the experiments reported in `Table 6`, the total algorithm runtime for 300 BO iterations is less than 60 seconds on the PD1 benchmark. In contrast, training and evaluating a neural network at each training epoch takes approximately 30 seconds for MobileNet, and up to 90–100 seconds for larger architectures like ResNet-50 or HRNet. This leads to a total training time of 9,000 to 30,000 seconds for the total training epochs (=300) spent during BO process, making the algorithmic overhead less than 0.7% (e.g., $0.7 \approx \frac{60}{9000} \times 100$), which is negligible.
>
> - We will include the above discussion in the revision to clarify the rationale for focusing on the number of BO evaluations.

---

> ### Author Response · Authors · 2025-08-06
>
> ---
>
> > [Q2-1] Regarding stopping criteria, I find your attempt to frame existing stopping mechanisms within your utility framework interesting. However, I do have two comments: I am not fully convinced that the mapping between the stopping mechanisms is as straightforward as suggested (and 1 to 1), particularly for [3] which operates under different assumptions about optimization and budget trade-offs. Moreover, given that the stopping criterion of CFBO is presented as one of your contributions, the paper would benefit from an explicit empirical comparison between CFBO's stopping mechanism and existing approaches (e.g., [3]). I do believe this would be helpful to show the advantages of your proposed method in practice while also making the connection to this related literature of early stopping BO.
>
> - We sincerely agree that the mapping between existing stopping mechanisms and our utility-based framework is not straightforward, and this is precisely why we stated that we were speaking "roughly" in the rebuttal.
>
> - As you correctly pointed out, the underlying assumptions of [3] differ significantly from ours. Specifically, [3] aims to **maximize true population performance**, whereas our focus—aligned with the standard practice in hyperparameter optimization—is to **maximize validation performance** as a proxy for generalization. In particular, [3] proposes a stopping rule based on estimating statistical error via cross-validation to ensure generalization to the test set, which is not directly accessible in our setup.
>
> - In contrast, our stopping criterion is explicitly designed to optimize **user utility over validation performance**, incorporating both performance and cost. This is in line with the typical setup of multi-fidelity BO, where the test performance (or population risk) is not available during training.
>
> - Furthermore, due to the nature of our benchmark datasets (LCBench, TaskSet, PD1), we are unable to apply cross-validation-based criteria as done in [3], since model checkpoints are not available.
>
> - In summary, while we sincerely agree that [3] is an **important and principled work** in the early stopping BO literature, **its goals and assumptions differ from ours, making a direct empirical comparison infeasible in our setting**. We will clarify this distinction and add a discussion of this in the revision.
>
> ---
>
> ### Reference
>
> [1] Rakotoarison, Herilalaina, et al. "In-context freeze-thaw bayesian optimization for hyperparameter optimization." ICML. 2024.
>
> [2] Kadra, Arlind, et al. "Scaling laws for hyperparameter optimization." NeurIPS. 2023.
>
> [3] Makarova, Anastasia, et al. "Automatic termination for hyperparameter optimization." International Conference on Automated Machine Learning. PMLR, 2022.

---

> > ### Comment · Reviewer_Ly5J · 2025-08-07
> > **Reply to Authors**
> >
> > Thank you for your responses.
> > Below, I would like to address a few remaining points regarding early stopping in BO.
> >
> > I do maintain that existing work on early stopping in BO, particularly for HPO, is highly relevant to your contribution (and I see it as positive that you are agreeing on this and will improve the presentation of this related work).
> > The downstream applications of your work and general early stopping in BO / HPO do overlap strongly.
> > In both cases, users want to early stop HPO, though for potentially different reasons.
> > In your approach, stopping occurs when expected performance improvement does not justify additional computational cost, while in other works such as [8], stopping is triggered when the noise in generalization error estimates dominates the potential improvements.
> > Both approaches ultimately result in saving computational resources.
> >
> > While I understand the challenges you have outlined, I disagree that direct empirical comparisons are completely unfeasible.
> > Your method could be compared against existing early stopping techniques to assess both final performance and computational cost savings.
> > Moreover, although [8] was not specifically designed for multi-fidelity HPO (and thus CFBO as a whole would likely have an advantage), they do discuss how to adapt their method for settings using a single holdout validation split.
> > I can see that your choice of benchmarks (LCBench, TaskSet, PD1) makes such comparisons more challenging, but not impossible.
> >
> > After considering the rebuttal, the other reviews, and the subsequent discussions, I find myself in a somewhat mixed position. While most of my initial concerns have been addressed, and I can acknowledge there is merit in your contributions, I remain reserved about advocating for acceptance.
> > I am willing to increase my score slightly, but overall the paper remains in a borderline state for me.

---

> ### Author Response · Authors · 2025-08-07
>
> We sincerely appreciate your active engagement, which we believe has meaningfully contributed to improving our work. Furthermore, we value the opportunity to engage with the thoughtful perspectives shared within the community. Please find our responses to your additional questions below:
>
> ---
>
> > I do maintain that existing work on early stopping in BO, particularly for HPO, is highly relevant to your contribution (and I see it as positive that you are agreeing on this and will improve the presentation of this related work). The downstream applications of your work and general early stopping in BO / HPO do overlap strongly. In both cases, users want to early stop HPO, though for potentially different reasons. In your approach, stopping occurs when expected performance improvement does not justify additional computational cost, while in other works such as [8], stopping is triggered when the noise in generalization error estimates dominates the potential improvements. Both approaches ultimately result in saving computational resources.
>
> - We sincerely agree that, although the underlying assumptions and exact goals differ, both approaches aim to mitigate computational inefficiency.
> - In particular, [8] offers an almost **hyperparameter-free automatic termination**, which is practically valuable for general HPO tasks.
> - Our work, however, targets a more specific (less explored) setting where users explicitly trade off computational cost and performance.
> - As you pointed out, we respectfully argue that our aim is to capture these "potentially different reasons"—such as how strongly to penalize computational cost or what form that penalty should take—by incorporating a user-defined utility function into the multi-fidelity HPO framework.
> - While there is strong conceptual overlap, we assure you that we will discuss the strengths of [8] (i.e., **principled and general applicability**), clarify the distinctions between the two approaches, and thoroughly include these discussions in the revision.

---

> ### Author Response · Authors · 2025-08-07
>
> ---
>
> > While I understand the challenges you have outlined, I disagree that direct empirical comparisons are completely unfeasible. Your method could be compared against existing early stopping techniques to assess both final performance and computational cost savings. Moreover, although [8] was not specifically designed for multi-fidelity HPO (and thus CFBO as a whole would likely have an advantage), they do discuss how to adapt their method for settings using a single holdout validation split. I can see that your choice of benchmarks (LCBench, TaskSet, PD1) makes such comparisons more challenging, but not impossible.
>
> - We sincerely agree that such comparisons are **challenging but not impossible**. Below, we respectfully explain what we have attempted to fairly compare [8] with CFBO and why [8] does not perform well in our setting:
>
>     1. We implemented [8] using FSBO, a transfer learning-based single-fidelity BO method, to ensure a fair comparison with CFBO in terms of transfer learning capabilities, and because [8] is originally designed for single-fidelity settings.
>     2. For computing `Eq. 8` in [8], we followed their suggestion in `Appendix A.2.3` and set $\beta_t = 2 \log\left(\frac{|\mathcal{X}| t^2 \pi^2}{6 \delta}\right)$, with $\delta = 0.1$ and $|\mathcal{X}|$ representing the number of hyperparameters.
>     3. As described in the last paragraph of `Section 3.2` in [8], we set the right-hand side of `Eq. 10` as a fixed threshold (e.g., 0.0001 as used in their Figure 1).
>     4. We further performed a grid search over the threshold (step size = 0.05) to maximize the given utility function using the meta-training split. The best thresholds were 3.95, 3.55, and 4.30 for LCBench, TaskSet, and PD1, respectively.
>     5. Using these thresholds, we applied the stopping rule from `Eq. 10` to FSBO during evaluation.
>
> - Despite these efforts, we unfortunately observed that the final utility values **decreased in most cases**, indicating inferior performance when integrating [8]. For example, the table below shows the normalized regrets under cost-sensitive HPO settings ($c=1, \alpha=2^{-5}$):
>
>     |Method|LCBench|TaskSet|PD1|
>     |-|-|-|-|
>     |FSBO|0.0386|0.0361|0.0193|
>     |FSBO+[8]|0.0426|0.0401|0.0212|
>     |CFBO|0.0287|0.0206|0.0136|
>
> - We believe this is due to several structural mismatches between [8] and our setting:
>     - Since FSBO does not support multi-fidelity evaluations, each selected configuration must be fully trained to the maximum budget (e.g., 50 epochs). Given a total budget of 300 epochs, this results in only 5–6 configurations being evaluated.
>
>      - Consequently, the key quantity $\bar{r}_t$, which depends on upper and lower confidence bounds of observed and unobserved performances—becomes unstable. The limited number of evaluations and the inherent noise in generalization error estimates make it difficult to obtain meaningful stopping signals in our experimental setting.
>
>     - Moreover, the absence of intermediate fidelities (e.g., partial training epochs) prevents the application of fine-grained early stopping strategies. This coarse granularity forces a trade-off: either terminate evaluations prematurely, potentially discarding promising configurations, or continue training to completion, which can lead to inefficient use of computational resources.
>
>     - We emphasize that these observations do not undermine the validity of [8] in its intended context. Rather, we believe that the observations show a misalignment between its assumptions and the constraints of our multi-fidelity, utility-driven setting. We will include this analysis and discussion in the revision for greater clarity.
>
> - In summary, although [8] is a principled and effective early stopping method in general HPO settings, we found that it does not translate well to **our multi-fidelity, utility-driven setup**. We will include this discussion in the revision.
>
> ---
>
> > After considering the rebuttal, the other reviews, and the subsequent discussions, I find myself in a somewhat mixed position. While most of my initial concerns have been addressed, and I can acknowledge there is merit in your contributions, I remain reserved about advocating for acceptance. I am willing to increase my score slightly, but overall the paper remains in a borderline state for me.
>
> - Thank you for clarifying your position. We also sincerely appreciate your willingness to increase your score. If our additional discussions have helped alleviate some of your remaining concerns, we hope this may be reflected in your overall evaluation.
>
> ---

---

> ### Author Response · Authors · 2025-08-08
>
> Dear Reviewer Ly5J,
>
> As the discussion period draws to a close (< 24 hours), we would like to express our sincere appreciation for your thoughtful engagement throughout.
>
> While we recognize that our empirical comparison with [8] may not fully resolve all of your concerns, we believe it offers valuable insights into the practical limitations of applying [8] within our setting—utility-driven, multi-fidelity HPO under constrained budgets, and an absence of cross-validation. These challenges create conditions under which existing early stopping methods may struggle, despite their strengths in broader HPO contexts.
>
> We assure you that **all points discussed during the rebuttal and discussion periods—including the strengths, assumptions, and applicability of [8]—will be thoroughly incorporated into the revision**. We believe this will help clarify our contributions, better position our method in the context of related work, and demonstrating the distinct challenges our setting presents.
>
> Best Regards,
>
> The Authors

---

> > ### Comment · Reviewer_Ly5J · 2025-08-09
> > **Response to Authors**
> >
> > Thank you for your follow up.
> > I will take your efforts in investigating FSBO+[8] into account when adjusting my final recommendation.
> > Also, thank you for being so pro-active and responsive during the rebuttal.

---

> > > ### Author Response · Authors · 2025-08-09
> > >
> > > We would rather take this opportunity to thank you for your active participation and constructive feedback, which have helped us clarify our position and improve the paper, and we apologize if our follow-up clarifications took too much of your time.
> > >
> > > Once again, we will expand the Related Work section (`Appendix A`) with a more substantial discussion on the literature of early-stopping BO, clarifying its benefits in terms of generalizability and principledness, and as well as its distinctions from our problem setup.
> > >
> > > Best Regards,
> > >
> > > The Authors

---

### Official Review · Reviewer_AJQp · 2025-06-30

**Clarity:** 3
**Significance:** 2
**Originality:** 3
**Rating:** 5
**Confidence:** 4

**Summary:**

The authors propose a cost-sensitive freeze-thaw BO approach that not only thaws the most promising configuration at each step, but also is aware of the cost of evaluating the thawed candidate. The awareness is realized by a utility function describing the trade-off between expected performance improvement and computational cost of the evaluation. An experimental evaluation shows that the proposed approach is competitive with other existing multi-fidelity approaches.

**Questions:**

see weaknesses

**Ethical Concerns:**

["NO or VERY MINOR ethics concerns only"]

**Limitations:**

The authors do not discuss the limitations of their approach. This should be added (at least in the Appendix). Also, there is no related work section in the paper (however, a "Background" section). Still, discussing some related work, such as FSBO and QuickTune, would be beneficial (can be done in the Appendix).

**Quality:**

3

**Strengths And Weaknesses:**

**Strengths**

- The work considers an important problem in BO
- The paper is written clearly
- The proposed approach is presented well, and the empirical evaluation is appropriate for the proposed method

**Weaknesses**

- The authors do not provide any theoretical considerations on their proposed utility score. Though not being essential, an analysis of the convergence of multi-fidelity BO using the proposed utility score would be interesting
- line 147-149: It is not clear to me what the authors mean by "selecting one point above" in this context.
- Choice of hyperparameters of utility function: I can imagine that, depending on the hyperparameter choice of the utility function, the BO might yield solutions that differ quite much in terms of their evaluation score. Can the authors provide best practices to set these parameters?
- Fig. 6: I'm wondering why QuickTune underperforms here. Since QuickTune comes with transfer capabilities, it would be nice if the authors could provide more details on that.
- Results on cost-sensitive MF-HPO: I'd expect FSBO to be better than CFBO because FSBO is not considering any trade-offs but merely performs for maximum performance. Therefore, I found it surprising to see that FSBO performs worse than CFBO throughout all tasks. Maybe the authors could comment on that.

---

> ### Author Rebuttal · Authors · 2025-07-30
>
> We sincerly appreciate you for your time and constructive comments which improve our paper. We respectfully address your concerns as following:
>
> ---
>
> >**[Q1]** The authors do not provide any theoretical considerations on their proposed utility score. Though not being essential, an analysis of the convergence of multi-fidelity BO using the proposed utility score would be interesting
>
> - We sincerely agree that theoretical analysis based on the utility function would further strengthen our work. While a full convergence analysis is beyond the scope of this paper, we note that our acquisition function is built on the classical **Expected Improvement (EI)** principle (`Eq. 3`), which is well-studied and underpins many convergence results in Bayesian optimization.
> - Our formulation generalizes EI by replacing the target (maximum performance) with a **utility function** that accounts for both performance and cost. This preserves the core idea of improvement in expectation while aligning optimization with user preferences.
> - Empirically, we show across diverse benchmarks and utility functions that CFBO leads to consistent and stable improvements over baselines (e.g., `Figure 4`, `Figure 6`), suggesting that the algorithm behaves robustly.
> - We view formal convergence guarantees under utility-based multi-fidelity BO as an important direction for future work and will mention this in the revision.
>
> ---
>
> >**[Q2]** line 147-149: It is not clear to me what the authors mean by "selecting one point above" in this context.
>
> - We sincerely apologize for the confusion. In `L141–150`, our intention is to illustrate how the user utility function can be approximated using the Bradley–Terry model with simulated preference data.
> - To fit the model, we construct preference pairs $(b, y_1)$ and $(b, y_2)$ and assign preference labels (e.g., $U(b, y_1) > U(b, y_2)$) based on a simulated user's behavior.
> - Specifically, we assume a user who prefers **any outcome that improves upon the baseline ifBO trajectory** in the black line of `Figure 2`. "selecting one point above" is simulated to reflect this assumption as follows:
> - For each budget $b$, let $y_b^{(\text{BO})}$ denote the performance of ifBO. We then sample $y_1 \sim \text{Uniform}(y_b^{(\text{BO})}, 1)$ and $y_2 \sim \text{Uniform}(0, y_b^{(\text{BO})})$, and record a preference $U(b, y_1) > U(b, y_2)$. This simulates **a user who consistently favors improvements over the ifBO trajectory at each budget level**.
> - We will clarify this explanation and notation in the revision.
>
> ---
>
> >**[Q3]** Choice of hyperparameters of utility function: I can imagine that, depending on the hyperparameter choice of the utility function, the BO might yield solutions that differ quite much in terms of their evaluation score. Can the authors provide best practices to set these parameters?
> - Thank you for pointing this out. We respectfully note that $c$ and $\alpha$ are not hyperparameters of the algorithm but **define the problem setup**: they encode the user's preference over the trade-off between model performance and computational cost.
> - In practice, users may not be able to specify this utility analytically. To address this, one can provide a small library of functional forms—e.g., linear ($c=1$), quadratic ($c=2$), square-root ($c=0.5$), or staircase—and then fit the utility using **preference learning** via the Bradley–Terry model. This is our recommended practice for estimating $c$ and $\alpha$ from a few preference comparisons.
> - Utility functions also allow for more tailored behavior. For example, if users have a hard minimum budget $B'$ they want to fully use, but prefer to stop early unless improvements are significant, their utility could be:
> $$
> U(b,\tilde{y}_b)=\begin{cases}\tilde{y}_b &\text{if }b \leq B',\\\ \tilde{y}_b - \alpha B'/B &\text{otherwise},\end{cases} \quad \text{where} \quad \alpha \rightarrow 1
> $$
> - This flexibility enables CFBO to reflect a wide range of practical user criteria. Moreover, experiments in `Figure 4` show CFBO performs well across diverse settings ($c \in \{1,2,0.5\}$ and $\alpha \in \{2^{-6},\ldots,2^{-2}\}$), demonstrating its robustness to different utility choices.
> - We will clarify the above in the revision.
>
> ---
>
> >**[Q4]** Fig. 6: I'm wondering why QuickTune underperforms here. Since QuickTune comes with transfer capabilities, it would be nice if the authors could provide more details on that.
> - Thank you for your insightful feedback. QuickTune$^\dagger$ (a transfer variant of DyHPO) performs comparably to non-transfer baselines, except on TaskSet. We attribute this to three key limitations: (1) a **simple deep kernel GP** surrogate [1] with limited expressivity, (2) its **greedy acquisition strategy**, and (3) the **lack of data augmentation**.
> - In contrast, CFBO uses a **Transformer-based PFN** [3] that scales better with data and supports richer transfer. As discussed in `Section 2.3`, we also apply a **mixup-style augmentation** to improve generalization across tasks.
> - Together with a **non-greedy acquisition** (when $\alpha=0$), these design choices allow CFBO to more effectively leverage transfer learning and outperform QuickTune on standard multi-fidelity HPO benchmarks.
> - Notably, FSBO—despite using the same limited architecture (deep kernel GP)—significantly performs better than QuickTune because it learns to predict only the final performance, avoiding greedy behavior.
> - We will include the above discussion in the revision.
>
>
> ---
>
> >**[Q5]** Results on cost-sensitive MF-HPO: I'd expect FSBO to be better than CFBO because FSBO is not considering any trade-offs but merely performs for maximum performance. Therefore, I found it surprising to see that FSBO performs worse than CFBO throughout all tasks. Maybe the authors could comment on that.
>
> - We kindly note that this is because **performance is measured based on user utility**, which accounts for both accuracy and cost.
> - As you noted, when utility considers only final performance (i.e., the conventional setting with $\alpha=0$), FSBO performs similarly to CFBO (`Figure 6`).
> - However, in the cost-sensitive setting ($\alpha>0$), FSBO ignores cost entirely and continues exploration even when the marginal utility gain is low. As a result, it achieves **lower overall utility** than CFBO, which actively trades off performance and cost.
> - We will clarify this distinction in the revision.
>
> ---
>
> ### References
>
> [1] Bradley, Ralph Allan, and Milton E. Terry. "Rank analysis of incomplete block designs: I. the method of paired comparisons." Biometrika. 1952.
>
> [2] Wilson, Andrew Gordon, et al. "Deep kernel learning." PMLR, 2016.
>
> [3] Rakotoarison, Herilalaina, et al. "In-context freeze-thaw bayesian optimization for hyperparameter optimization." ICML. 2024.

---

> > ### Comment · Reviewer_AJQp · 2025-08-05
> >
> > I thank the authors for their extensive rebuttal, my concerns have been addressed accordingly and I will keep my score.

---

> > > ### Author Response · Authors · 2025-08-06
> > >
> > > We sincerely appreciate your decision and assure you that we will clarify the above discussion in the revision.

---

### Official Review · Reviewer_uqhd · 2025-07-03

**Clarity:** 3
**Significance:** 2
**Originality:** 2
**Rating:** 4
**Confidence:** 4

**Summary:**

This paper considers a Bayesian optimization method that combines freeze-thaw BO, preference learning, and transfer learning. The method uses a variant of expected improvement on the preference function (utility), where the incumbent threshold is set to the most recent utility, rather than the previously evaluated point with max utility. The method maximizes EI over the parameters and target epoch. The method leverages a regret-based stopping criterion for decide when to stop the entire HPO loop. The paper shows how training the PFN via transfer learning (via mixup) improves performance.

**Questions:**

* The intro discusses the motivation being that budget is typically limited and users may want to allocate more budget to training the model than to exploration. Why not focus the paper on BO algorithms that explicitly account for a known budget and plan accordingly (using lookahead and planning w.r.t the budget)?
* Is it reasonable for a BO user to pick the parametric form of their utility function? That seem hard for a user to specify
* How does performance change if parts of the AF are ablated: a) if only final performance is optimized for rather than the optimizing the target epoch, b) if the threshold used in EI is the best incumbent and not the most recent?

**Ethical Concerns:**

["NO or VERY MINOR ethics concerns only"]

**Final Justification:**

Thanks for the response and clarifying the ablations. Most of my concerns have been addressed, and I have increased my score. My main criticism is lack of originality, as the paper combines many existing approaches. Nevertheless, the combination is effective and tackles an important problem.

**Limitations:**

yes

**Quality:**

2

**Strengths And Weaknesses:**

Strengths
* The empirical performance is strong
* Sensitivity analyses show the best choice of HPs for for the Beta CDF in the stopping criterion

Weaknesses
* The empirical evaluation gives the proposed method a sizable advantage notably since methods are evaluated w.r.t  the utility which the proposed method is aware of (it omnisciently knows the right preference function class) and baselines are not.
* The contribution is pretty small as the paper combines freeze-thaw BO, transfer learning, and preference learning.
* The method introduces quite a few hyperparameters (HPs for the Beta CDF, choice of parametric function class, etc)
* The transfer learning method requires retraining the PFN for each new use case

---

> ### Author Rebuttal · Authors · 2025-07-30
>
> We sincerly appreciate you for your time and constructive comments which improve our paper. We respectfully address your concerns as following:
>
> ---
> > **[Q1]** Baseline unfairness.
> - Thank you for pointing this out. We already provide a fair comparison under the conventional multi‑fidelity HPO setting with $\alpha=0$ (`Figure 6`). In this case the utility collapses to pure performance, so CFBO and all baselines optimize the same objective. CFBO still achieves superior results, showing its advantage is **not tied to privileged knowledge**.
> - Furthermore, incorporating our utility into existing baselines is **fundamentally difficult** because $U$ depends on future performance–cost trade‑offs and requires a probabilistic inference over full learning curves. DyHPO [1] predicts only the next step, and FSBO [2] predicts only the final performance, so they **cannot evaluate expected future utility** or apply our adaptive stopping and acquisition rules.
> - For fairness, we tuned a fixed stopping threshold ($\delta = 0.2$ in `Eq. 4`) for all baselines using learning curve (LC) training dataset. They cannot use the adaptive criterion $\delta_b$ in `Eq. 5` because it relies on computing the future utility in `Eq. 6`.
> - ifBO [3] is the closest prior work because it models long‑horizon performance, but it lacks transfer learning and the utility‑aware criteria. We therefore built **CFBO‑NT**, which augments ifBO with our acquisition and stopping rules while excluding transfer learning. CFBO‑NT still outperforms all baselines on cost‑sensitive benchmarks, confirming that the observed gains come from **our methodological contributions**.
> - We will discuss the above details in the revision.
> ---
> > **[Q2]** Limited novelty.
> - We respectfully disagree with this assessment. We acknowledge that each individual ingredient, freeze–thaw BO, transfer learning across tasks, and preference learning, has appeared in prior work. The novelty of CFBO lies in **how these elements are brought together to tackle a problem** that, to our knowledge, has not been formally addressed before: hyperparameter optimization when **users explicitly trade off model performance against computational cost**.
> - As discussed in `L32‑36`, for example, cloud‑service users often operate under credit budgets. They care about final performance, but they also place a penalty on the HPO cost so that credits remain for other jobs. Existing algorithms either **ignore cost entirely or treat it only as a hard budget constraint**; they do not optimise expected utility that mixes both factors. Empirically, we show (see the answer to **[Q1]**) that traditional methods are ineffective in this setting.
> - CFBO is designed for this realistic scenario. We formulate a utility function encoding user preference, identify the need for probabilistic inference to evaluate the utility, and integrate freeze–thaw BO (for curve extrapolation) with transfer learning (for sample efficiency) to realise the objective. Thus our contribution consists of
>     1. introducing a practical, cost‑aware HPO **problem formulation**,
>     2. defining a **concrete utility‑based objective**,
>     3. analysing why **existing methods fall short**, and
>     4. presenting CFBO, **an effective solution** that—while built from known components—solves the new problem end‑to‑end.
> - The advance therefore also resides in the problem definition and its principled resolution, not merely in assembling familiar techniques.
> - We will clarify the above in the revision.
> ---
> > **[Q3]** Hyperparameters.
> - We kindly note that CFBO introduces **only one tunable hyperparameter**, $\beta$; all other quantities are either fixed for fairness or specified by the task itself.
> - The adaptive stopping rule uses $\text{BetaCDF}$ with parameters $\beta$ and $\gamma$. To align CFBO with baselines, we select a fixed stopping threshold $\delta = 0.2$ using the learning‑curve training set, then set $\gamma = \log_{0.5} 0.2$ so that the adaptive rule reproduces this threshold when $p_b = 0.5$. Since $\gamma$ is fixed, it does not introduce additional tuning.
> - The coefficients $c$ and $\alpha$ reflect how users trade off cost and performance. These are **task parameters**, not algorithm hyperparameters, and apply equally to all methods.
> - `Figure 7(d)` shows CFBO’s performance is robust for $\beta \ge e^{-1}$, allowing users to select a reasonable default without costly sweeps.
> - We will clarify this distinction and the limited tuning burden in the revision.
> ---
> > **[Q4]** PFN retraining required.
> - Thank you for pointing this out. We acknowledge this concern and have already evaluated a **non‑transfer variant**, CFBO‑NT, described in `L276–278`. CFBO‑NT integrates ifBO with our utility‑aware acquisition function (`Eq. 3`) and adaptive stopping criterion (`Eq. 5`) but omits the PFN retraining step.
> - CFBO‑NT still achieves substantial improvements over all baselines that do not use transfer learning, demonstrating that the core ideas of CFBO—utility‑aware acquisition and adaptive stopping—remain effective without a pre‑trained PFN.
> - Practitioners lacking a learning‑curve training dataset can therefore adopt CFBO‑NT to obtain most of CFBO’s benefits without the overhead of PFN retraining.
> ---
> > **[Q5]** Why not plan for a fixed budget?
> - We respectfully believe this is a misunderstanding. In our formulation the user **need not** spend the entire budget; they may stop BO early and reserve the remaining credits for downstream training or other jobs. So the choice is whether **the extra benefit from more exploration** is worth the credits that could instead be spent on other tasks.
> - A hard‑budget BO that plans to exhaust the budget can be sub‑optimal in this setting. If the performance gains flatten out well before the cap, it wastes credits that could yield higher overall utility elsewhere. By contrast, CFBO casts the trade‑off as a **continuous utility function** that penalizes additional cost; the algorithm stops automatically once expected gains fall below that penalty.
> - We evaluate CFBO across a wide range of cost‑sensitivity levels ($\alpha$) and cost‑penalty shapes ($c$) to show that it consistently outperforms methods that ignore costs and use a fixed stopping rule (`Figure 4`). When $\alpha$ is large—mimicking a tight, known budget—CFBO’s decisions converge to those of a hard‑budget BO (e.g., $\alpha=2^{-2}$ in `Figure 5`), demonstrating that our framework **subsumes** the fixed‑budget case while remaining effective when early stopping is beneficial.
> - Practitioners who truly face a rigid, must‑spend budget can still apply CFBO by setting $\alpha=0$ until the must-spend budget and apply $\alpha>0$ after the budget; the method will naturally plan to use the budget. In all other cases, CFBO offers higher expected utility by adapting spending to the observed learning progress.
> - We will include the above discussion in the revision.
> ---
> > **[Q6]** Users may not define a utility function.
> - We sincerly agree with that most users cannot write down an exact analytic utility.
> - To address this, we embed **preference learning** via the Bradley–Terry model [4] (`L133–140`). The system presents pairs of outcomes $(b_1, y_1)$ and $(b_2, y_2)$; the user simply chooses the one they prefer, e.g., $U(b_1, y_1)>U(b_2, y_2)$, and the model fits parameters that explain those choices.
> - This approach assumes only that the user can (1) state a preference between two options and (2) select a rough shape—linear ($c=1$), quadratic ($c=2$), square-root ($c=0.5$) or staircase—from a small predefined library. These shapes cover a broad range of cost‑sensitivity patterns while keeping inference tractable.
> - In `L141–150` we demonstrate a user who prefers any trajectory that outperforms the baseline ifBO trajectory at every point. We simulate preference pairs under this assumption, and `Figure 2` shows that the fitted utility closely follows the true trend.
> - Furthermore, as noted in `L149–150`, `Figure 9` in `Appendix B` shows that with about 30 pairwise comparisons the learned utility already tracks the true curve closely; with 100 comparisons the approximation error is negligible. The interaction burden is therefore modest.
> - Once the utility parameters are estimated, CFBO proceeds unchanged, allowing users who cannot specify a utility function explicitly to benefit from the method with minimal preference feedback.
> ---
> > **[Q7]** What if acquisition function is ablated? a) optimize only the final performance, b) use the best incumbent as the EI threshold
> - We kindly note that the requested ablations coincide with CFBO evaluated under the **conventional HPO setting**, where the cost‑penalty weight is set to $\alpha = 0$.
> - With $\alpha = 0$ the utility simplifies to $U(b,\tilde{y}_b) = \tilde{y}_b$, so CFBO’s acquisition reduces to maximizing predicted **final performance** while ignoring computational cost—exactly ablation (a).
> - Under the same simplification, the threshold $U_p$ in `Eq. 3` becomes the **current best incumbent** because the utility input is the pair $(b, \tilde{y}_b)$, where $\tilde{y}_b$ is the running best performance. This reproduces ablation (b).
> - Empirically, CFBO with $\alpha = 0$ is reported in `Figure 6`; it still outperforms all baselines in this pure‑performance regime, demonstrating that the performance gains are not contingent on the cost‑aware components.
> ---
> ### References
>
> [1] Wistuba, Martin, Arlind Kadra, and Josif Grabocka. "Supervising the multi-fidelity race of hyperparameter configurations." NeurIPS. 2022.
>
> [2] Wistuba, Martin, and Josif Grabocka. "Few-shot Bayesian optimization with deep kernel surrogates." arXiv. 2021.
>
> [3] Rakotoarison, Herilalaina, et al. "In-context freeze-thaw bayesian optimization for hyperparameter optimization." ICML. 2024.
>
> [4] Bradley, Ralph Allan, and Milton E. Terry. "Rank analysis of incomplete block designs: I. the method of paired comparisons." Biometrika. 1952.

---

> ### Comment · Reviewer_uqhd · 2025-08-05
>
> Thanks for the response and clarifying the ablations. Most of my concerns have been addressed, and I'll increase my score.
>
> > So the choice is whether the extra benefit from more exploration is worth the credits that could instead be spent on other tasks.
>
> This requires that the utility function captures preferences over spending on other future tasks, which might be hard for the user to estimate. That said, using pairwise feedback is a reasonable approach.
>
>
> > CFBO‑NT still achieves substantial improvements over all baselines that do not use transfer learning
>
> My point was that there are faster methods for transfer learning that re-training a PFN, which could be quite expensive. Although CFBO-NT performs quite well, FSBO often outperforms it and I would imagine is much cheaper than CFBO when including retraining time.
>
>
> > Why not plan for a fixed budget?
>
> This question was largely aimed to understand how this method compares against a method that seeks to optimally use a pre-specified budget. I.e. does stopping early help, compared to optimally using a fixed budget? The latter bakes in a user-preferred allocation for this task vs other future tasks, but I do understand that a user may not know the potential gains a priori for a given task, and so using stopping rules can be quite preferable.

---

> ### Author Response · Authors · 2025-08-06
>
> We sincerely appreciate you for **your efforts, participation on this discussion period, and positive decision**. We address your additional concerns below:
>
> ---
>
> > "So the choice is whether the extra benefit from more exploration is worth the credits that could instead be spent on other tasks." This requires that the utility function captures preferences over spending on other future tasks, which might be hard for the user to estimate. That said, using pairwise feedback is a reasonable approach.
>
> - Thank you for pointing this out. Our focus is solely on the present task, where we model user preferences using the Bradley–Terry model with pairwise feedback. Therefore, we fully agree that it is nearly infeasible to design a utility function that captures preferences over spending on other future tasks.
>
> - Incorporating user preferences over future tasks into the utility function would be an interesting direction for future work. We will include this discussion in the revision.
>
> ---
>
> > "CFBO‑NT still achieves substantial improvements over all baselines that do not use transfer learning". My point was that there are faster methods for transfer learning that re-training a PFN, which could be quite expensive. Although CFBO-NT performs quite well, FSBO often outperforms it and I would imagine is much cheaper than CFBO when including retraining time.
>
> - We sincerely agree with your point. We train the LC extrapolator of CFBO on only training data, while not using any synthetic data as done in PFNs. Our main focus is to demonstrate the way to maximize user utility, and the way to leverage synthetic data or pre-trained PFN may be underexplored comparatively.
> - To improve this direction, one can consider incorporate the recent advance in the PFNs fine-tuning literature, such as Tune-Tables [1] which proposes prompt-tuning for real data (not synthetic) for TabularPFNs.
> - However, we believe this is out of scope in our paper in that our main focus is user utility. We will discuss it in the revision.
>
> ---
>
> > "Why not plan for a fixed budget?" This question was largely aimed to understand how this method compares against a method that seeks to optimally use a pre-specified budget. I.e. does stopping early help, compared to optimally using a fixed budget? The latter bakes in a user-preferred allocation for this task vs other future tasks, but I do understand that a user may not know the potential gains a priori for a given task, and so using stopping rules can be quite preferable.
>
> - We sincerely appreciate your understanding. We respectfully argue that stopping early can lead to higher user utility compared to fully consuming a fixed budget. By the definition of user utility, if the performance improvement from additional evaluations is marginal, the utility can decrease due to increased cost.
>
> - In this respect, optimally using a fixed budget is equivalent to stopping at the point in the BO trajectory where the maximum utility is achieved. `Figure 8` illustrates this comparison, showing the improvements of BO methods when applying **optimal early stopping** (i.e., stopping at the utility peak along the trajectory). Indeed, we observe consistent gains across all methods under this optimal policy.
>
> - We hope the above explanation clarifies our position, and we will revise the manuscript to better emphasize this comparison in the final version.
>
> ---

---

### Note · Authors · 2025-08-11

Dear AC and all reviewers,

We take this final remark as a valuable opportunity to **express our sincere gratitude for your effort** in overseeing and reviewing our paper. We would also like to outline how, upon acceptance, we will incorporate the concerns addressed during the rebuttal and discussion period into the revision:

---
>`Section 1`: Introduction
- **Problem setups (uqhd and Ly5J)**: We will add a figure illustrating our novel problem setup and differences from existing HPO literature.
---
>`Section 2`: Approach
- **Notation pointer (Ly5J)**: We will add a pointer in `L103` to the notation table.
- **PFN clarification (Vxb6)**: We will add a pointer in `L125` to `Appendix D` for more details on PFN.
- **Utility estimation (uqhd, AJQp, and Vxb6)**: We will clarify how we estimate the user utility in `L127–150`.
- **Practice for utility (AJQp)**: We will add recommendations for determining user utility in `L127–150`.
- **Acquisition function (AJQp and Ly5J)**: We will expand `L154–164` to better convey the principled nature of the acquisition function.
- **PFN retraining (uqhd)**: We will add a discussion in `L248–250` on efficiently retraining PFN with recent advances.
---
>`Section 3`: Experiments
- **Baseline fairness (uqhd and Ly5J)**: We will emphasize the fairness of baseline comparisons with CFBO-NT in `L278` and `L295–298`.
- **Hyperparameter clarification (uqhd, AJQp, and Ly5J)**: We will clarify task ($c, \alpha$) and stopping criterion hyperparameters ($\delta, \gamma$), and the maximum budget ($B$), along with our settings, in `L289–298`.
- **Experimental results analysis (AJQp)**: We will extend the analysis of QuickTune and FSBO in `L311–321` and `L322–336`.
- **Runtime analysis (Ly5J)**: We will add a paragraph after `L336` discussing runtime analysis and its negligible impact, with `Table 5` in `Appendix E`.
- **Visual clarity (Vxb6)**: We will improve the visual clarity of `Figure 4 and 6` by utilizing the increased page limits.
---
>`Section 4`: Conclusion
- **Limitations (Ly5J)**: We will add a paragraph after `L399` discussing the limitations of pool-based HPO with PFNs.
---
>`Appendix`
- **Early stopping BO (Ly5J)**: We will add several paragraphs in `Appendix A` summarizing our rebuttal and discussion-period insights on early stopping BO, connecting them to our framework.
- **Notation table (Ly5J)**: We will introduce a new `Appendix B` containing and explaining the full notation table.
---

Best Regards,

The Authors

---

### Decision · Program_Chairs · 2025-09-17

**Decision:**

Accept (poster)

**Comment:**

This work builds on freeze-thaw Bayesian optimization, developing a cost-sensitive version that early stops non-promising hyperparameter configurations. The authors propose to leverage an utility function that is a function of performance and cost, which they use in conjunction with a dedicated acquisition function and a regret-based stopping criterion. The authors provided solid empirical evidence to support the main claims of the paper.

The authors provided an extensive rebuttal, addressing the questions and concerns raised by the reviewers. The authors also captured the key improvements to incorporate into the final version of the manuscript. The work is timely and the authors addressed runtime concerns in a satisfactory manner by pointing to the negligible overhead compared to model training time.

While the work builds on a combination of existing approaches, it does so in an effective and novel way as acknowledged by most reviewers. One reviewer remained more critical, indicating that the early stopping literature was not well represented in the original submission. The authors provided additional data points during the rebuttal to fill that gap. On balance, I am in agreement with one of the reviewers indicating that the paper has included enough relevant baselines to warrant acceptance.